# Towards Poisoning Fair Representations

**Tianci Liu[1], Haoyu Wang[1], Feijie Wu[1], Hengtong Zhang[2], Pan Li[3], Lu Su[1], Jing Gao[1]**
[1]Purdue University    [2]Tencent AI Lab    [3]Georgia Institute of Technology
[1]{liu3351,wang5346,wu1977,lusu,jinggao}@purdue.edu
[2]htzhang.work@gmail.com    [3]panli@gatech.edu

## Abstract

Fair machine learning seeks to mitigate model prediction bias against certain demographic subgroups such as elder and female. Recently, fair representation learning (FRL) trained by deep neural networks has demonstrated superior performance, whereby representations containing no demographic information are inferred from the data and then used as the input to classification or other downstream tasks. Despite the development of FRL methods, their vulnerability under data poisoning attack, a popular protocol to benchmark model robustness under adversarial scenarios, is under-explored. Data poisoning attacks have been developed for classical fair machine learning methods which incorporate fairness constraints into shallow-model classifiers. Nonetheless, these attacks fall short in FRL due to notably different fairness goals and model architectures. This work proposes the first data poisoning framework attacking FRL. We induce the model to output unfair representations that contain as much demographic information as possible by injecting carefully crafted poisoning samples into the training data. This attack entails a prohibitive bilevel optimization, wherefore an effective approximated solution is proposed. A theoretical analysis on the needed number of poisoning samples is derived and sheds light on defending against the attack. Experiments on benchmark fairness datasets and state-of-the-art fair representation learning models demonstrate the superiority of our attack.

## 1 Introduction

Machine learning algorithms have been thriving in high-stake applications such as credit risk analysis. Despite their impressive performance, many algorithms suffered from the so-called *fairness* issue, i.e., they were shown *biased* against under-represented demographic subgroups such as *females* in decision making (Barocas & Selbst, 2016; Buolamwini & Gebru, 2018). As remedies, fair machine learning has accumulated a vast literature that proposes various fairness notions and attempts to achieve them (Pedreshi et al., 2008; Dwork et al., 2011; Hardt et al., 2016). For *shallow* models such as logistic regression and support vector machine, fairness notions are often defined upon *scalar* model predictions and have been well-studied, see Kamishima et al. (2011); Zafar et al. (2017) and reference therein for instances. We refer to this family of work as *classical fair machine learning*. Recently, fair representation learning (FRL) with deep neural networks (DNNs) has attracted great attention (Xie et al., 2017; Madras et al., 2018; Creager et al., 2019; Sarhan et al., 2020). FRL learns *high-dimensional* representations for downstream tasks that contain *minimal information* of sensitive features (i.e., the memberships of demographic subgroups). These information-based fairness notions equip FRL with higher transferability than classical methods (Creager et al., 2019).

Despite the success of fair machine learning methods, not much is known about their vulnerability under data poisoning attacks until very recent studies (Chang et al., 2020; Solans et al., 2021; Mehrabi et al., 2021). Data poisoning attacks aim to maliciously control a model's behaviour to achieve some attack goal by injecting *poisoning samples* to its training data and are widely used to benchmark the robustness of machine learning models in adversarial scenarios (Bard & Falk, 1982; Biggio et al., 2012). Recently researchers successfully attacked classical fair machine learning methods such as fair logistic regression (Mehrabi et al., 2021) and exacerbated bias in model predictions, thereby hurting fairness. But it is still an open question whether FRL suffers from a similar threat[1].

---

[1]See App A for a more detailed discussion on attacking fair representations versus downstream classifiers.

Notwithstanding, devising a poisoning attack to degrade fairness of representations proves to be a non-trivial task. The difficulty is from two aspects. First, high-dimensional representations are more complicated to evaluate fairness than scalar predictions in classical fair machine learning. This makes existing attack goals for fairness degradation against the latter fall short to apply. Secondly, fair representations implicitly depend on the training data, and non-convex DNNs make this dependency hard to control by previous optimization- or heuristic-based attacks on classical fair machine learning. Optimization-based attacks (Solans et al., 2021) need the victim model to be simple enough (e.g., convex) to analyze. Heuristics such as label flipping (Mehrabi et al., 2021) do not directly optimize the attack goal, thereby often requiring great effort to design good manipulations to success.

We propose the first data poisoning attack that directly targets on fairness of high-dimensional representations as shown in Figure 1. Our attack is optimization-based with a new attack goal. Specifically, following the common principle behind FRL that ideal fair representations should contain no information of sensitive features (Moyer et al., 2018; Zhao et al., 2019; Creager et al., 2019), we devise our attack goal to maximize the mutual information (MI) between the learned representations and sensitive features. The attack goal entails a bilevel optimization, whose challenges lie in two-folds.

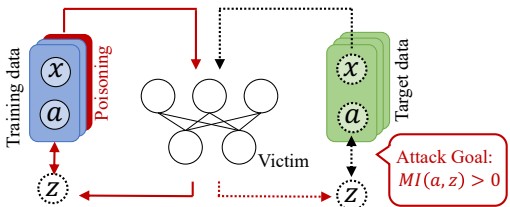

Figure 1: Framework of data poisoning attack (in red) on FRL. Irrelevant components such as class labels are omitted. The attacker poisons training data to contaminate the victim training (solid lines), resulting in unfair representations $z$ for target data (dotted lines) such that its MI to sensitive feature $a$ is maximized. The MI is supposed to be small before the attack.

Firstly, MI does not admit analytic form, making its maximization challenging. Instead, a variational lower bound of MI (Barber & Agakov, 2003) inspires us to use the negative loss of the optimal classifier that predicts sensitive features from representations as a proxy. To avoid the complexity of training such a classifier in the inner loop of the optimization, we further connect the classification loss with a principled notion of *data separability*. FLD scores give us an analytic measure of such and simplify the optimization substantially. Notably, FLD score as a valid data separability measure does not rely on the mixture-of-Gaussian assumption (Fisher, 1936), and our empirical results also show that this assumption is not essential for our purpose in practice. To our best knowledge, *we propose the first attack goal based on MI explicitly towards fair machine learning that is highly principled and flexible. We also analyze its connection to demographic parity, one of the most popular fairness notions.* This is one of the most significant contributions of our work.

Secondly, representations are learned by DNNs, whose dependency on poisoning samples is impossible to track exactly. Consequently, one cannot identify training on what poisoning samples the victim will produce the most fairness-violating representations. We solve this problem approximately by matching the upper- and lower-level gradients following Geiping et al. (2020): The gradient of the victim from FLD score gives the desired update direction; when it matches the gradients of the victim on poisoning samples, training the victim on poisoning samples will solve the attack goal approximately. To improve stealthiness of the attack on fairness degradation in our scenarios, we design better constraints on how much poisoning samples can deviate from clean samples. In specific, we define a more proper valid range based on the data type, and an elastic-net penalty (Chen et al., 2018) to encourage such deviation to be sparse. Meanwhile, *we derive the first theoretical minimal number of poisoning samples required by gradient-matching based attacks (Geiping et al., 2020) to succeed under regular conditions. This bound is crucial for practical attacks, as using more poisoning samples will increase the chance of being detected (Geiping et al., 2020; Koh et al., 2022).* This theoretical analysis is another contribution of this paper and can be of independent interest to users of all gradient-matching based attacks.

Extensive experimental results in Section 4 demonstrate high effectiveness of our attacks on four representative FRL methods using as few as 5% of training data for poisoning. In the remaining part of this paper, we review related works in Section 5, and conclude the work in Section 6.

## 2 PROPOSED METHOD

We propose the first *white-box clean-label* data poisoning attack on deep learning-based fair representation learning (FRL) to degrade fairness. We follow previous attacks on classical fair machine learning (Chang et al., 2020; Mehrabi et al., 2021) and suppose a worst-case threat model, which has full knowledge and control of the victim model, as well as the access to part of the training samples.

### 2.1 PRELIMINARIES

We denote a dataset consisting of $N$ datapoints by $\mathcal{D} = \{\boldsymbol{x}_i, a_i, y_i\}_{i=1}^N$, where $\boldsymbol{x}_i \in \mathbb{R}^M$ represents a multivariate nonsensitive feature, $a_i \in \{0, 1\}$ is a binary sensitive feature, and $y_i$ is a binary class label. As adopted in literature (e.g., Moyer et al. (2018); Reddy et al. (2021)), we assume $\boldsymbol{x}$ has continuous values that can be fed into neural networks. A FRL model parameterized by $\theta$ has an encoder $h$ to learn representation from the nonsensitive feature by $\boldsymbol{z}(\theta) = h(\boldsymbol{x}; \theta)$ and is trained on $\mathcal{D}$ to minimize some fairness-aware loss $L(\mathcal{D}; \theta)$. A data poisoning attack aims to maliciously control the training of this victim model by *perturbing* a few training samples to minimize some attack loss $U(\theta)$. The perturbed training samples are referred to as *poisoning samples*.

**Mutual information (MI)-based fairness** aims to minimize $I(a, \boldsymbol{z})$ between sensitive feature $a$ and representation $\boldsymbol{z}$. It makes fair representations highly transferable on a wide range of downstream classification tasks and has become the *de facto* principle in FRL (Moyer et al., 2018; Creager et al., 2019; Zhao et al., 2019). Formally, the data processing inequality (Cover, 1999) states that $I(a, \boldsymbol{z}) \geq I(a, g(\boldsymbol{z})) \geq 0$ holds for any classifier $g$ acting on $\boldsymbol{z}$. In addition, $I(a, g(\boldsymbol{z})) = 0$ is equivalent to demographic parity (DP, Zemel et al. (2013)), one of the most popular group fairness notions. At a colloquial level, *if representations from different demographic groups are similar to each other, then any classifier acting on the representations will be agnostics to the sensitive feature and fair thereof.* This condition remains valid without access to $y$ when DP cannot be evaluated.

### 2.2 POISONING FAIR REPRESENTATIONS FORMULATION

Motivated by the importance of MI-based fairness in FRL, we attack the fairness on some target data $\mathcal{D}_{\text{ta}}$ by maximizing $I(a, \boldsymbol{z})$. This involves a bilevel optimization problem. Given a victim $\theta$ and its lower-level loss $L(\mathcal{D}; \theta)$, the training data $\mathcal{D} = \mathcal{D}_{\text{po}} \cup \mathcal{D}_{\text{cl}}$ consists of $P$ poisoning samples $\mathcal{D}_{\text{po}}$ and clean samples $\mathcal{D}_{\text{cl}}$, the attacker wants to minimize $-I(a, \boldsymbol{z})$ over target data $\mathcal{D}_{\text{ta}}$ by learning perturbations $\Delta = \{\delta_p\}_{p=1}^P$ to add on $\mathcal{D}_{\text{po}}$ through solving

$$\min_{\Delta \in \mathcal{C}} -I(a, \boldsymbol{z}(\theta^*(\Delta))), \quad \text{s.t. } \theta^*(\Delta) = \text{argmin}_\theta L(\mathcal{D}_{\text{po}}(\Delta) \cup \mathcal{D}_{\text{cl}}; \theta). \quad (1)$$

Our *clean-label attack* leaves original label $y$ unpoisoned and only perturbs nonsensitive feature $\boldsymbol{x}$, i.e., $\mathcal{D}_{\text{po}}(\Delta) = \{\boldsymbol{x}_p + \delta_p, a_p, y_p\}_{p=1}^P$, under constraint $\mathcal{C}$ which will be detailed shortly.

**Connection to attacking group fairness.** Our attack is principled and jeopardizes DP. We use well-defined metric variation of information (VI, Kraskov et al. (2005)). For $y$ and $a$, their VI distance is $VI(y, a) = H(y) + H(a) - 2I(y, a)$ where $H(\cdot)$ is the entropy. Applying triangle inequality to $g(\boldsymbol{z})$, $y$, and $a$ gives us $I(g(\boldsymbol{z}), a) \geq I(g(\boldsymbol{z}), y) + I(a, y) - H(y) = I(g(\boldsymbol{z}), y) - H(y \mid a)$. By maximizing $I(\boldsymbol{z}, a)$ that upper bounds $I(g(\boldsymbol{z}), a)$, a successful attack diminishes the guarantee for the MI-based fairness. When $H(y \mid a) < I(g(\boldsymbol{z}), y)$, fitting $g$ to predict $y$ from $\boldsymbol{z}$ that maximizes $I(g(\boldsymbol{z}), y)$ will force $I(g(\boldsymbol{z}), a)$ to increase, thereby exacerbating the DP violation. Notably, $H(y \mid a)$ depicts how dependent $y$ and $a$ are and is expected small when an fairness issue exists (Hardt et al., 2016). We provide empirical success of attacking DP with our attack goal in Appendix E.3.

Unfortunately, Eq. (1) is intractable with difficulty lying in *finding the most fairness-violating representations* and *the method for acquiring them*. Mathematically, the first problem involves $I(a, \boldsymbol{z})$ which lacks an analytic expression. This makes its computation non-trivial, let alone its maximization. The second entails a feasible set in the lower-level optimization that is NP-hard to identify due to the non-convexity of deep neural networks. We solve Eq. (1) approximately as follows.

### 2.3 UPPER-LEVEL APPROXIMATION: FISHER'S LINEAR DISCRIMINANT (FLD)

The lack of analytic form for $I(a, \boldsymbol{z})$ necessitates some approximations. Our first step is to lower bound MI by a negative binary-cross-entropy (BCE) loss which is easier to estimate. For any classi-

fier $g$ that predicts $a$ from $\boldsymbol{z}$, let $q(a \mid \boldsymbol{z})$ be the distribution learned by the optimal $g^*$, we have

$$I(a, \boldsymbol{z}) = \mathbb{E}_{p(a, \boldsymbol{z})} \left[ \log \frac{p(a|\boldsymbol{z}) q(a|\boldsymbol{z})}{p(a) q(a|\boldsymbol{z})} \right] \overset{(a)}{\geq} \mathbb{E}_p \left[ \log \frac{q(a|\boldsymbol{z})}{p(a)} \right] = \mathbb{E}_p \left[ \log q(a|\boldsymbol{z}) \right] + \mathbb{E}_p \left[ -\log p(a) \right],$$

where the inequality $(a)$ holds from omitting some non-negative KL terms and is tight when $g^*$ recovers the true distribution $p(a \mid \boldsymbol{z})$ as shown in Barber & Agakov (2003). On the other hand, since $\mathbb{E}_p [-\log p(a)] = H(a) \geq 0$, the first term, negative BCE loss of $g^*$, is a lower bound for $I(a, \boldsymbol{z})$, and is a measure of how fair the representations are (Feng et al., 2019; Gupta et al., 2021). We dub the BCE loss of $g^*$ the *optimal BCE loss*.

However, substituting the MI maximization from Eq. (1) with minimizing the optimal BCE loss does not make it solvable: $g^*$ depends on $\boldsymbol{z}$'s for $\mathcal{D}_{\text{ta}}$ and how to update them to minimize the BCE loss of $g^*$ is unknown. This requires differentiate through the whole optimization procedure.

To walk around this challenge, we note that the optimal BCE loss of $g^*$ measures how *difficult* to *separate* $\boldsymbol{z}$ for $\mathcal{D}_{\text{ta}}$ with $a = 0$ or $1$. While this difficulty is hard to tackle directly, it can be approximated by the *data separability*: If the two classes of data are more separable, one can expect the classification simpler and the optimal BCE loss lower. Motivated by this, we instead maximize Fisher's linear discriminant (FLD) score, a closed-formed data separability measure.

Specifically, suppose the two classes of representations have mean $\boldsymbol{\mu}^0, \boldsymbol{\mu}^1$ and covariance $\mathbf{S}^0, \mathbf{S}^1$ respectively. FLD maps them to a 1-dimensional space via linear transformation $\boldsymbol{v}$ which induces *separation* $s_{\boldsymbol{v}} = (\boldsymbol{v}^\top \boldsymbol{\mu}^0 - \boldsymbol{v}^\top \boldsymbol{\mu}^1)^2 / (\boldsymbol{v}^\top \mathbf{S}^0 \boldsymbol{v} + \boldsymbol{v}^\top \mathbf{S}^1 \boldsymbol{v})$. Its maximal value over $\boldsymbol{v}$, termed as *FLD score*, is $s = (\boldsymbol{\mu}^0 - \boldsymbol{\mu}^1)^\top (\mathbf{S}^0 + \mathbf{S}^1)^{-1} (\boldsymbol{\mu}^0 - \boldsymbol{\mu}^1)$ when $\boldsymbol{v} \propto (\mathbf{S}^0 + \mathbf{S}^1)^{-1} (\boldsymbol{\mu}^0 - \boldsymbol{\mu}^1)$. This equation allows us to compute its gradient with respect to $\boldsymbol{z}$ for $\mathcal{D}_{\text{ta}}$, which gives the direction to update these representations in order to make it less fair. For stability we regularize $s$ by

$$s = (\boldsymbol{\mu}^0 - \boldsymbol{\mu}^1)^\top (\mathbf{S}^0 + \mathbf{S}^1 + c\mathbf{I})^{-1} (\boldsymbol{\mu}^0 - \boldsymbol{\mu}^1), \tag{2}$$

and resort to solving the following bilevel optimization:

$$\min_{\Delta \in \mathcal{C}} -s(\theta^*(\Delta)), \quad \text{s.t. } \theta^*(\Delta) = \operatorname{argmin}_\theta L(\mathcal{D}_{\text{po}}(\Delta) \cup \mathcal{D}_{\text{cl}}; \theta). \tag{3}$$

In Appendix B we extend our attack to multi-class sensitive feature scenarios.

*Remark* 2.1. Maximizing $I(\boldsymbol{z}, a)$ is a general framework to poison FRL and admits other proxies such as sliced mutual information (Chen et al., 2022), kernel canonical correlation analysis (Akaho, 2007), and non-parametric dependence measures (Székely et al., 2007). In this work, we use FLD because of its conceptual simplicity, interpretability, and good empirical performance. As one may recognize, when $p(\boldsymbol{z} \mid a = 1)$ and $p(\boldsymbol{z} \mid a = 0)$ are Gaussian with equal variance, FLD is optimal (Hamsici & Martinez, 2008) whose BCE loss attains the tight lower bound of $I(\boldsymbol{z}, a)$ up to constant $H(a)$. In this case, our method provably optimize the lower bound of $I(\boldsymbol{z}, a)$ whereas other proxies may not due to the lack of direct connections to mutual information. While the Gaussianity may not hold in general, FLD score as a measure of data separability is still valid (Fisher, 1936), and we verify its efficacy for our goal in Appendix E.2, where we show that FLD score is highly informative for the empirical minimal BCE loss of a logistic regression.

## 2.4 Lower-level Approximation: Elastic-Net GradMatch (ENG)

Bilevel optimization in Eq. (3) enjoys a tractable upper-level loss but its lower-level optimization remains challenging due to the use of deep neural networks in FRL and needs further approximation. To this end, we fix parameter $\theta$ in a *pre-trained* victim model and treat its lower-level gradient on poisoning samples $\nabla_\theta L(\mathcal{D}_{\text{po}}(\Delta); \theta)$ as a function of $\Delta$. An attack is launched by aligning $\nabla_\theta L(\mathcal{D}_{\text{po}}(\Delta); \theta)$ and $-\nabla_\theta s(\theta)$ by maximizing their cosine similarity

$$\mathcal{B}(\theta, \Delta) = -\frac{\langle \nabla_\theta s(\theta), \nabla_\theta L(\mathcal{D}_{\text{po}}(\Delta); \theta) \rangle}{\|\nabla_\theta s(\theta)\| \|\nabla_\theta L(\mathcal{D}_{\text{po}}(\Delta); \theta)\|}, \tag{4}$$

with respect to $\Delta$. When the two directions are matched, gradient descents on the lower-level loss $L$ from poisoning samples will decrease the upper-level loss $-s$ as well.

The concept of gradient matching in poisoning attacks was initially introduced in GradMatch by Geiping et al. (2020) for image classification. To enhance the stealthiness of such attacks in fairness degradation scenarios, we impose two new constraints on the learning perturbations. First, a

*hard* constraint $\mathcal{C}$ mandates that poisoning samples must reside within the clean data domain, i.e., $\min_{\boldsymbol{x}_c \in \mathcal{D}_{cl}} x_{cm} \leq \delta_{pm} + x_{pm} \leq \max_{\boldsymbol{x}_c \in \mathcal{D}_{cl}} x_{cm}$ for all feature dimensions ($1 \leq m \leq M$) and poisoning samples ($1 \leq p \leq P$), this allows using dimension-specific ranges. Second, the *soft* constraint employs the elastic-net penalty (Zou & Hastie, 2005; Chen et al., 2018) to promote sparsity in each $\delta_p$ (i.e, only a few dimensions are perturbed). Combining these constraints, we formulate below optimization problem aimed at poisoning fair representations:

$$\min_{\Delta \in \mathcal{C}} -\mathcal{B}(\theta, \Delta) + \sum_{p=1}^{P} \left( \lambda_1 \|\delta_p\|_1 + \lambda_2 \|\delta_p\|_2^2 \right). \tag{5}$$

In execution, we rescale the three terms before tuning $\lambda$'s to avoid bearing with the magnitude difference. Non-differentiable $L_1$ norms are tackled with the iterative shrinkage-thresholding algorithm (ISTA, Beck & Teboulle (2009)) $\delta_p^{k+1} = \mathrm{Proj}_{\mathcal{C}} \left( S_{\lambda_1} \left[ \delta_p^k + \alpha^k \nabla(\mathcal{B}(\theta, \Delta) - \lambda_2 \|\delta_p\|_2^2) \right] \right)$, where $S_{\lambda_1}$ is the element-wise projected shrinkage-thresholding function and $\mathrm{Proj}_{\mathcal{C}}$ is the projection onto $\mathcal{C}$. We refer to our attack by **E**lastic-**N**et **G**radMatch (ENG) and summarize it in Algorithm 1.

**Computation complexity.** Denote the number and dimension of poisoning samples by $P$ and $M$, iteration numbers of running the attack by $T$, dimension of $\theta$ by $D$. The computation complexity of ENG and GradMatch (Geiping et al., 2020) are $\mathcal{O}(TM + TDPM)$ and $\mathcal{O}(TDPM)$, respectively. The additional computation cost $\mathcal{O}(TM)$ due to the one-step ISTA in ENG is marginal.

## 3 ANALYSIS ON MINIMAL NUMBER OF POISONING SAMPLES

It is non-trivial to minimize the number of poisoning samples that deteriorate the victim model's performance. An insufficient amount may not impact lower-level optimization, while a large amount makes it easy to detect the attack, leading to a direct failure (Huang et al., 2020; Koh et al., 2022). Our analysis is built upon the convergence of upper-level loss $U(\theta) \equiv -s(\theta)$. Our conclusion relies on the following assumptions.

**Assumption 3.1** (smooth upper- and lower-level losses). There exists $C > 0$ such that upper-level loss $U(\theta)$ and lower-level loss $L(\theta)$ are $C$-smooth, i.e., their gradients are $C$-Lipschitz continuous.

**Assumption 3.2** (attack well-trained victims only). Before attack, the victim is well-trained so its gradient on each clean sample performs like a random noise with mean zero and finite norm $\sigma$.

**Assumption 3.3** (well-matched gradients). After being attacked, gradient $\nabla_\theta L(\theta)$ evaluated on any poisoning sample is an unbiased estimator of the gradients $\nabla_\theta U(\theta)$ with bounded error norm, i.e., for any poisoning sample $p$, $\nabla L_p(\theta) = \nabla_\theta U(\theta) + \epsilon_p$ where $\mathbb{E}[\epsilon_p] = 0$ and $\mathbb{E}[\|\epsilon_p\|] \leq \sigma$.

Assumption 3.1 is a fairly weak condition that has been widely used (Colson et al., 2007; Mei & Zhu, 2015; Sinha et al., 2017), the rest two are introduced by us and are well-suited to the context of our proposed framework. Assumption 3.2 is valid because $\theta$ undergoes thorough training with the clean samples. Assumption 3.3 presumes that given constant gradient $\nabla_\theta U(\theta)$, we can construct an unbiased estimator with $P$ poisoning samples $\mathbb{E}[\frac{1}{P} \sum_{p=1}^{P} \nabla_\theta L_p(\theta)] = \nabla_\theta U(\theta)$. With these assumptions, we obtain the following theorem.

**Theorem 3.4.** *Suppose that Assumption 3.1, 3.2, and 3.3 hold. Let $P$ and $N$ be the number of poisoning and total training samples, respectively. Set the learning rate to $\alpha$ and the batch size to $n$. Then, the ratio of poisoning data $P/N$ should satisfy*

$$\frac{P}{N} \geq c + \frac{\alpha C \sigma^2}{2n \|\nabla_\theta U(\theta)\|^2} + \frac{\alpha C}{2}, \tag{6}$$

*such that the upper-level loss $U(\theta)$ is asymptotic to an optimal model. Here $c$ is a small constant (e.g., $10^{-4}$) for sufficient descent, and $\theta$ is a well-trained model on the clean data.*

Deferring the proof to Appendix C, we summarize the underlying idea. We assume the pretrained victim converged with respect to $\nabla_\theta L(\theta)$. Besides, after applying ENG, a poisoning sample can (in expectation) induce a gradient equal to $\nabla_\theta U(\theta)$ so training the victim model on it will optimize the upper-level loss $U(\theta)$. However, clean samples may obfuscate this goal as their lower-level gradients do not depend on the upper-level loss. To counteract with this effect, a minimal portion of poisoning

samples are needed to dominate the training. In practice, learning rate $\alpha$ is often small compared with $C$, batch size $n$ is large, and $\|\nabla_\theta U(\theta)\|$ is much greater than 0. Therefore, the minimal portion bound is expected to be smaller than 1.

**Practical Significance.** Theorem 3.4 sheds light on the difficulty of ENG and other GradMatch-based attacks, whereon a defending strategy can be built. If batch size $n$ is large, the attack is simpler and needs fewer poisoning samples. So reducing $n$ should help defense and we evaluate its performance in Appendix E.7. In addition, term $\sigma^2$ is affected by the lower-level gradients, whose increase will require more poisoning samples. This helps explain why adding noises to gradients can defend against GradMatch as verified in Geiping et al. (2020).

---

**Algorithm 1** Craft Poisoning Samples with ENG Attack

1: **Input:** clean data $\mathcal{D}_{cl}$, poisoning data $\mathcal{D}_{po}$, target data $\mathcal{D}_{ta}$; victim $\theta$ and its lower-level loss $L$; number of pre-training epochs $E$ and attack iterations $T$.
2: Fix $\mathcal{D}_{po}$ unperturbed. Pretrain victim on $\mathcal{D}_{po} \cup \mathcal{D}_{cl}$ for $E$ epochs and obtain $\theta^E$.
3: Compute upper-level gradient $\nabla_\theta U(\mathcal{D}_{ta}; \theta^E)$.
4: Randomly initialize $\Delta^0 = \{\delta_p, p = 1, \ldots, P\}$ in $\mathcal{C}$.
5: **for** $t = 1, \ldots, T$ **do**
6:     Compute lower-level gradient $\nabla_\theta L(\mathcal{D}_{po}(\Delta^t); \theta^E)$ as a function of $\Delta^t$.
7:     Compute ENG loss in (5) and update $\Delta^t$ with ISTA.
8: **end for**
9: **return** $\mathcal{D}_{po}$ when using $\Delta^T$.

---

# 4 EXPERIMENTS

We evaluate the proposed attack on four FRL models trained on two fairness benchmark datasets and show its effectiveness through extensive experiments. Ablation studies and practical insights are also given to help understand how the attack succeeds in poisoning victims.

## 4.1 SETUP

**Attack Goals.** We consider three variants of Eq. (2) to maximize: (a) FLD follows Eq. (2) with $c = 10^{-4}$ to stabilize the covariance estimation. (b) sFLD takes the same form as FLD but does not back-propagate through covariance terms when computing $-\nabla_\theta s(\theta)$. (c) EUC replaces the covariance terms with an identity matrix[2].

**Attacks.** An attack to maximize score X using ENG is referred to as ENG-X, for instance, ENG-FLD is to maximize FLD. Conceptually, ENG-EUC is suitable for small $\mathcal{D}_{ta}$ as it omits covariance matrix, which can be unstable to estimate. ENG-FLD should be favored when $\mathcal{D}_{ta}$ is large as it is based on the exact FLD score $s$ and should measure separability more accurately, which is expected to benefit solving Eq. (5) as well. ENG-sFLD strikes a balance in between by using only part of covariance information.

**FRL Victims.** We select four representative FRL models as our victims. CFAIR and CFAIR-EO (Zhao et al., 2019) are based on adversarial learning and seek to achieve different fairness notions. Non-adversarial learning based ICVAE-S and ICVAE-US (Moyer et al., 2018) differ in whether $y$ is accessible. We follow the official codes to implement the victims as detailed in Appendix D.

**Datasets.** We train victims on two benchmark datasets from UCI repository that are extensively studied in fair machine learning, which are pre-processed[3] following Zhao et al. (2019); Reddy et al. (2021). Adult (Kohavi, 1996) contains 48,842 samples of US census data with 112 features and the objective is to predict whether an individual's yearly income is greater than $50K dollars or not. The sensitive feature is *gender*. German (Dua & Graff, 2017) consists of 1,000 samples of personal financial data with 62 features and the objective is to predict whether or not a client has a good credit score. The sensitive feature is binarized *age* as in Moyer et al. (2018) and we adjust the threshold to increase its correlation with label[4]. In Appendix E.6 we study multi-class sensitive feature *race*. On both datasets, we leave 20% of total samples out as $\mathcal{D}_{ta}$. More results on COMPAS (Dieterich et al., 2016) and Drug Consumption Datasets (Dua & Graff, 2017) are presented in Appendix. E.8.

---

[2]This score is equivalent to the squared Euclidean distance between $\boldsymbol{\mu}^0$ and $\boldsymbol{\mu}^1$ and gets its name thereof.

[3]In Appendix E.1 we discuss the practicability of perturbing pre-processed versus raw data

[4]We increase the lower threshold of defining advantaged group from 25 to 30.

**Evaluation.** We treat decrease in the BCE loss of a logistic regression predicting $a$ from $z$ as a measure of increase in $I(z, a)$. To verify how group fairness and representation utility can be affected, we present the exacerbation of DP violation and accuracy of predicting $y$ from $z$ in Appendix E.3 and Appendix E.4 respectively. Representations are extracted after training the victims on poisoned Adult and German dataset for 20 and 50 more epochs respectively considering their size difference.

**Baselines.** We compare ENG-based attacks with four variants of anchor attack (AA), a recent heuristic generic poisoning attack on classical fair machine learning (Mehrabi et al., 2021). RAA-y and RAA-a randomly picks one training sample from the subgroup with $(y = 1, a = 0)$ and one with $(y = 0, a = 1)$ after each epoch, then makes copies of the two chosen samples with flipped $y$ or $a$ respectively. NRAA-y and NRAA-a replaces the random selection in RAA counterparts with picking from each subgroup the training sample that has the most neighbors within a pre-specified radius and has not been selected yet. (N)RAA-y were proposed in Mehrabi et al. (2021), and we implement (N)RAA-a following the same logic. Note that (N)RAA-y are not clean-label, (N)RAA-a are but they directly modify the sensitive feature. In contrast, our attacks only modify the nonsensitive feature. Moreover, all baselines are allowed to use *any* training sample to poison, while ours can only perturb a *given* randomly selected set of poisoning samples. *These differences put our proposed attacks in an unfavorable situation under direct comparison. Nevertheless, ours can still outperform the four baselines by a large margin.*

## 4.2 COMPARISON BETWEEN ENG AND (N)RAA.

We compare three ENG-based attacks with penalty coefficients $\lambda_1 = 0.0025, \lambda_2 = 0.005$ (Eq. (5)) against four AA attacks under different settings where 5% to 15% of training data are used for poisoning. Performance of an attack is measured by the decrease of BCE loss, higher is better, and corresponding DP violations are reported in Appendix E.3.

Figure 2 shows results averaged over 5 replications. Three ENG-based attacks achieved notable performance (on both BCE loss and DP violations) in various settings. In contrast, AA encountered severe failures, for instance, when attacking CFAIR trained on German dataset, only RAA-a succeeded with all the three budgets. Such failures cannot be fully attributed to the budgets: (N)RAA-y succeeded with budget 10% but failed with budget 15%. (N)RAA-a occasionally achieved the best performance because of much stronger capacities. Nonetheless, the proposed ENG-based attacks beat AA baselines in terms of better and more reliable performance by a large margin in most case. When comparing the three proposed attacks with each other, their performance difference matched our previous analysis, e.g., ENG-FLD gave the best result on larger Adult dataset. These results clearly establish the efficacy of our attack.

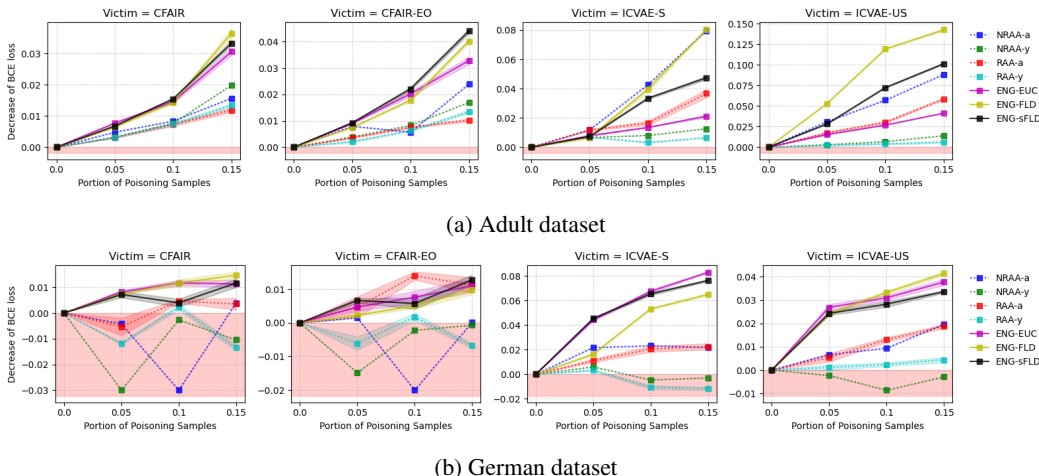

(a) Adult dataset

(b) German dataset

Figure 2: ENG-based attacks reduce BCE loss more than AA baselines with less portion of poisoning samples. Results are averaged over 5 independent replications and bands show standard errors.

### 4.3 PERFORMANCE OF ELASTIC-NET PENALTY

Next, we sprovide a sensitivity analysis of ENG-based attacks against the choices of $\lambda_1$ and $\lambda_2$. We set $\lambda_2 = 2\lambda_1$ and vary $\lambda_1$ following Chen et al. (2018). Results are again averaged over 5 replications. Figure 3a exhibits how elastic-net penalty affects the performance of ENG-FLD under different poisoning budgets where victims are trained on Adult dataset. More results are deferred to Appendix E.5 due to page length but conclusions here hold in general. All three attacks are relatively robust to small and intermediate level of elastic-net penalty. Moreover, improvement from applying the elastic-net penalty is observed (e.g., column 1 in Fig 3a). This implies that the penalty can actually help stabilize the optimization.

We further study how elastic-net penalty regularizes the learned perturbations by computing $L_1$ and $L_2$ norms. We only present $L_1$ norms of perturbations using ENG-EUC attack on Adult dataset as an illustration and defer others to Appendix E.5 after observing similar trends. According to Figure 3b, $L_1$ norms were significantly shrank by elastic-net penalty with mild degradation on the attack performance. For instance, elastic-net penalty with $\lambda_1 = 0.0025$ effectively reduced the $L_1$ norm of perturbations by a third without hurting the performance of attacking CFAIR. When a more stealthy attack is wanted, $\lambda_1 = 0.01$ was able to launch a satisfactory attack with only one half budget of perturbation norm. These results clearly show the efficacy of our proposed ENG-based attacks.

Given these merits of elastic-net penalty, one may ask if it can be used in other attacks such as AA. In Appendix E.5 we discuss the difficulty of doing so and highlight its affinity to our attack by nature.

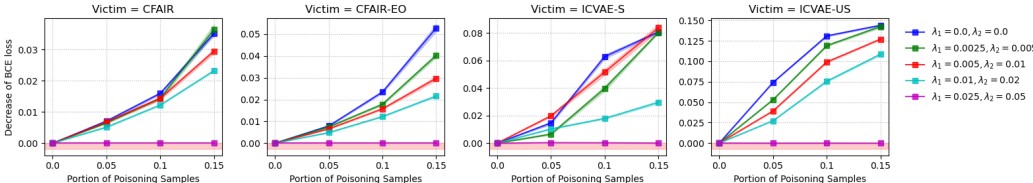

(a) Decrease of BCE loss is insensitive to under small and intermediate level of elastic-net penalty.

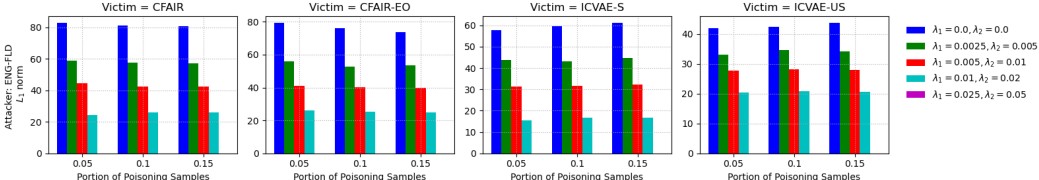

(b) $L_1$ norm of learned perturbations is effectively restricted by elastic-net penalty.

Figure 3: Decrease of BCE loss and $L_1$ norm of perturbations learned by ENG-FLD attack. Victims are trained on Adult dataset and results are averaged over 5 replications.

### 4.4 ROBUST FEATURES OF VICTIMS

ENG-based attacks can do feature selections, and we consider *unselected* features as *robust* ones for the victim, in the sense that they did not help in attacking. We end this section with a case study on this aspect. Table 1 shows the percentage of selected features and corresponding robust features when attacking ICVAE-S and ICVAE-US trained on Adult dataset with ENG-FLD. From the table, elastic-net penalty successfully reduced the perturbed features by 20%, and robust features for ICVAE-US and ICVAE-S are largely overlapped. These results helped understand how ENG-FLD was launched by walking around these robust features when attacking the victims. Note that this robust features identification is not applicable for AA-based attacks.

## 5 RELATED WORK

**Fair representation learning (FRL)** is a family of algorithms to learn fair representations from nonsensitive features such that any downstream tasks (e.g., classification) acting on these representations will be fair. Towards this goal, different approaches to removing sensitive information from

learned representations have been proposed. To name a few, Zemel et al. (2013) set representations to be multinomial and used fairness violations based on the representations as penalties, and Madras et al. (2018) derived different bounds for these violations used for adversarial regularization. Mutual information and other measures between distributions have also been used as penalties to encourage independence between the representation and sensitive features, either in an adversarial (Xie et al., 2017; Creager et al., 2019) or non-adversarial way (Louizos et al., 2015; Moyer et al., 2018; Sarhan et al., 2020; Wang et al., 2021). Recently, Zhao et al. (2019) proposed to learn different encoders on different label groups with theoretical guarantees and achieved state-of-the-art performance in an adversarial manner Reddy et al. (2021). Moyer et al. (2018), on the other hand, provided a non-adversarial based solution and also got promising results. We select representative methods from adversarial and non-adversarial regimes to test our attack.

**Data poisoning attack** aims to achieve some attack goal with a victim model by controlling its training via injecting poisoned samples. Early works showed that simple heuristics such as label flipping (Barreno et al., 2010; Paudice et al., 2019) can succeed in attack. However, these poisoning samples often look unnatural and are easy to detect (Papernot & McDaniel, 2018; Paudice et al., 2019). Consequently, clean-label attacks that only modify the poisoning samples' features but not labels are preferred (Shafahi et al., 2018).

Table 1: Percentage of selected features and top robust features from Adult dataset, $P/N$ denotes the portion of poisoning samples, attacker is ENG-FLD.

| $P/N$ | $\lambda_1$ ($\lambda_2 = 2\lambda_1$) | | | | Top Robust Features |
|---|---|---|---|---|---|
| | 0 | 0.0025 | 0.005 | 0.01 | |
| ICVAE-S 0.05 | 0.99 | 0.96 | 0.89 | 0.73 | *workclass, marital-status,* |
| 0.1 | 0.99 | 0.95 | 0.89 | 0.75 | *occupation, native-country* |
| 0.15 | 0.99 | 0.97 | 0.89 | 0.77 | |
| ICVAE-US 0.05 | 0.95 | 0.89 | 0.84 | 0.79 | *native-country,* |
| 0.1 | 0.95 | 0.91 | 0.85 | 0.79 | *occupation, workclass,* |
| 0.15 | 0.96 | 0.91 | 0.85 | 0.79 | *education* |

Another drawback of heuristic attacks is the lack of performance guarantee as they do not directly solve the attack goals. In practice they may perform less effective.

Bilevel optimization is widely used for data poisoning attacks (Bard & Falk, 1982; Biggio et al., 2012; Geiping et al., 2020; Jagielski et al., 2021; Koh et al., 2022). For convex victims such as logistic regression and support vector machine, the lower-level optimization is characterized by the KKT condition. This reduces the bilevel optimization to a constrained optimization that can be solved exactly (Mei & Zhu, 2015). For other victims, unfortunately, when their optimal solutions are NP-hard to identify, so are the bilevel optimization problems (Colson et al., 2007; Sinha et al., 2017) and inexact solutions are needed. When the second-order derivatives of the lower-level loss is cheap, using influence function to identify influential samples for the victim training and poisoning them can produce strong attacks (Koh & Liang, 2017). These attacks have been successfully applied to classical fair machine learning (Chang et al., 2020; Solans et al., 2021; Mehrabi et al., 2021), but the non-convexity of neural networks and expensive influence function computations make them unsuitable for poisoning FRL. Approximate solutions for attacking deep learning models have been proposed recently. For instance, inspired by model-agnostic meta-learning (MAML, Finn et al. (2017)), MetaPoison (Huang et al., 2020) back-propagated through a few unrolled gradient descent steps to capture dependency between the upper- and lower-level optimizations. GradMatch (Geiping et al., 2020) matched the gradient of upper- and lower-level losses and achieved state-of-the-art performance. However, it is unclear how to apply them to poison FRL. In this work, we propose the first work towards this goal and reduce it to an approximate optimization that can be handled.

## 6 CONCLUSION AND FUTURE WORKS

We develop the first data poisoning attack against FRL methods. Driven by MI-based fairness in FRL, we propose a new MI maximization attack goal and reveal its connection to existing fairness notion such as demographic parity. We derive an effective approximate solution to achieve this attack goal. Our attack outperforms baselines by a large margin and raises an alert of the vulnerability of existing FRL methods. We also theoretically analyze the difficulty of launching such an attack and establish an early success of principled defense. Motivated by promising results on tabular data, which is the primary focus of many FRL methods (Moyer et al., 2018; Zhao et al., 2019), we plan to extend our attack to fair machine learning on large-scale image and text datasets that also relies on deep neural networks, and delve into attacking these methods in the future. In addition, Jagielski et al. (2021) showed that an attack can be much more effective towards certain subpopulations and impossible to defend against, and we plan to explore this for further improvement of our attack.

## ACKNOWLEDGEMENT

This work is supported in part by the US National Science Foundation under grant NSF IIS-2226108. Any opinions, findings, and conclusions or recommendations expressed in this material are those of the author(s) and do not necessarily reflect the views of the National Science Foundation.

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

## A    MOTIVATION OF ATTACKING FAIR REPRESENTATIONS

Attacking mutual information at representation level, as formalized in Section 2, aligns better with fair representation learning (FRL) and is more universal than attacking classifiers. For example, a bank may want to obtain a representation $z$ for each user that can be used to determine their eligibility for *existing* and *upcoming* financial products such as credit cards without concerning about fairness again (Zhao et al., 2019). Here each financial product is associated with a (unique) label, and determining the eligibility entails a classification task. In this case, there are two challenges for delivering a classification-based attack. First, one has to determine and justify which classifier to use and why consider the fairness of this specific classification task. Second, for any upcoming financial product, its label does not exist and one cannot obtain classifier (we need this label to train the classifier), let along attacking it. In contrast, a representation-level attack can overcome the two challenges in a single shot. As discussed in Section 2, for any classifier $g$ acting on $z$, by maximizing the mutual information $I(z, a)$ between $z$ and sensitive feature $a$, $I(g(z), a)$ will be maximized so long as the fairness concern exists. This implies launching attack on all labels simultaneously, including the ones where classifiers cannot be trained.

## B    EXTENSION TO MULTI-CLASS SENSITIVE FEATURE

To attack a FRL method trained on multi-class sensitive feature $a \in [K]$, We first define $\tilde{a}_k = \mathbf{1}(a = k)$ to mark whether the sample belongs to the $k$-th sensitive group. Then we immediately have $I(z, a) \geq I(z, \tilde{a}_k)$ thanks to the data processing inequality. This implies that

$$I(z, a) \geq \frac{1}{K} \sum_{k=1}^{K} I(z, \tilde{a}_k).$$

Note that in RHS each term is the mutual information between $z$ and binarized $\tilde{a}_k$ and is lower bounded by the BCE loss, therefore we can approximate each one with a FLD score readily. This idea is similar to transforming a $K$-class classification to $K$ one-vs-all binary classifications. We report empirical results in Appendix E.6.

Further simplification is viable by choosing some specific $k$ and attacking $I(z, \tilde{a}_k)$ only. This allows one to launch an attack when $K$ is large. However, it is noteworthy that handling a sensitive feature that involves many groups (large $K$) is a general challenge for fair machine learning. The difficulty is twofold. First, it involves more complicated constraints, making the problem harder to optimize. Second, by categorizing data into more fine-grained sensitive groups, each group will have fewer samples and the algorithm may suffer from unstable estimation issues (Jiang et al., 2022; Liu et al., 2023). As a result, when the number of sensitive groups is large, fair machine learning methods often bin them into a few larger groups in pre-processing (Zemel et al., 2013; Moyer et al., 2018; Zhao et al., 2019; Reddy et al., 2021).

## C    OMITTED PROOF OF THEOREM 3.4

We start with restating Theorem 3.4.

**Theorem C.1.** *Suppose that Assumption 3.1, 3.2, and 3.3 hold. Let $P$ and $N$ be the number of poisoning and total training samples, respectively. Set the learning rate to $\alpha$, and assign the batch size with $n$. Then, the ratio of poisoning data (i.e., $P/N$) should satisfy*

$$\frac{P}{N} \geq c + \frac{\alpha C \sigma^2}{2n \|\nabla_\theta U(\theta)\|^2} + \frac{\alpha C}{2}, \tag{7}$$

*such that the upper-level loss $U(\theta)$ (i.e., negative FLD score) is asymptotic to an optimal model. Here $c$ is a small constant (e.g., $10^{-4}$) for sufficient descent, and $\theta$ is a well-trained model on the clean data.*

*Proof.* Without loss of generality, we assume that the victim is trained by mini-batched stochastic gradient descent, i.e., given current parameter $\theta^{\text{old}}$, the new value is updated by

$$\theta^{\text{new}} = \theta^{\text{old}} - \frac{\alpha}{n} \sum_{i=1}^{n} \nabla_\theta L_i(\theta^{\text{old}}),$$

where $L_i(\theta)$ denotes the loss on the $i$-th training sample. Let $p$ denote the number of poisoning samples selected in the current batch, we have that $p$ follows a Hypergometric distribution. Given $p = n_p$ poisoning samples in the current batch, we collect all randomness in the minibatch gradient as

$$e^{\text{old}} = \sum_{i=1}^{n_p} \epsilon_i + \sum_{i=1}^{n-n_p} \nabla_\theta L_i(\theta^{\text{old}}).$$

According to Assumption 3.3 and 3.2 we have $\mathbb{E}[e^{\text{old}}] = 0$ and

$$
\begin{aligned}
\mathbb{E}\left[\|e^{\text{old}}\|^2\right] &= \mathbb{E}\left[\left\|\sum_{i=1}^{n_p} \epsilon_i + \sum_{i=1}^{n-n_p} \nabla_\theta L_i(\theta^{\text{old}})\right\|^2\right] \\
&= \sum_{i=1}^{n_p} \mathbb{E}[\|\epsilon_i\|^2] + \sum_{i=1}^{n-n_p} \mathbb{E}[\|\nabla_\theta L_i(\theta^{\text{old}})\|^2] + 0 \\
&\leq n\sigma^2,
\end{aligned}
$$

as all crossing terms have mean zero. Moreover, we can express

$$
\begin{aligned}
\sum_{i=1}^{n} \nabla_\theta L_i(\theta^{\text{old}}) &= \underbrace{\sum_{i=1}^{n_p} \nabla_\theta L_i(\theta^{\text{old}})}_{\text{Poisoning samples}} + \underbrace{\sum_{i=1}^{n-n_p} \nabla_\theta L_i(\theta^{\text{old}})}_{\text{Clean samples}} \\
&= \sum_{i=1}^{n_p} \nabla_\theta U(\theta^{\text{old}}) + \sum_{i=1}^{n_p} \epsilon_i + \sum_{i=1}^{n-n_p} \nabla_\theta L_i(\theta^{\text{old}}) \\
&= n_p \nabla_\theta U(\theta^{\text{old}}) + e^{\text{old}}.
\end{aligned}
$$

This implies that further given Assumption 3.2, the minibatch gradient is a biased estimator of the gradient of the upper-level loss. To see this, by the law of total expectation

$$
\begin{aligned}
\mathbb{E}\left[\frac{1}{n}\sum_{i=1}^{n} \nabla_\theta L_i(\theta^{\text{old}})\right] &= \mathbb{E}\left[\mathbb{E}\left[\frac{1}{n}\sum_{i=1}^{n} \nabla_\theta L_i(\theta^{\text{old}}) \mid p = n_p\right]\right] \\
&= \frac{1}{n}\mathbb{E}\left[\mathbb{E}\left[p\nabla_\theta U(\theta^{\text{old}}) + e^{\text{old}} \mid p = n_p\right]\right] \\
&= \mathbb{E}\left[p\nabla_\theta U(\theta^{\text{old}})\right] \\
&= \frac{1}{n}\nabla_\theta U(\theta^{\text{old}})\mathbb{E}[p] \\
&= \frac{P}{N}\nabla_\theta U(\theta^{\text{old}}).
\end{aligned}
$$

Under Assumption 3.1, descent lemma gives us

$$
\begin{aligned}
U(\theta^{\text{new}}) &\leq U(\theta^{\text{old}}) + \langle \nabla_\theta U(\theta^{\text{old}}), \theta^{\text{new}} - \theta^{\text{old}}\rangle + \frac{C}{2}\|\theta^{\text{new}} - \theta^{\text{old}}\|^2 \\
&= U(\theta^{\text{old}}) - \alpha\langle \nabla_\theta U(\theta^{\text{old}}), \frac{1}{n}\sum_{i=1}^{n}\nabla_\theta L_i(\theta^{\text{old}})\rangle + \frac{\alpha^2 C}{2n^2}\|\sum_{i=1}^{n}\nabla_\theta L_i(\theta^{\text{old}})\|^2.
\end{aligned}
$$

Take expectation on both side with respect to the mini-batch we get

$$\mathbb{E}[U(\theta^{\text{new}})] \leq U(\theta^{\text{old}}) - \alpha\langle\nabla_\theta U(\theta^{\text{old}}), \frac{P}{N}\nabla_\theta U(\theta^{\text{old}})\rangle + \frac{\alpha^2 C}{2n^2}\mathbb{E}\|\sum_{i=1}^n \nabla_\theta L_i(\theta^{\text{old}})\|^2$$

$$= U(\theta^{\text{old}}) - \frac{\alpha P}{N}\|\nabla_\theta U(\theta^{\text{old}})\|^2 + \frac{\alpha^2 C}{2n^2}\mathbb{E}\|\sum_{i=1}^n \nabla_\theta L_i(\theta^{\text{old}})\|^2, \tag{8}$$

and

$$\mathbb{E}\|\sum_{i=1}^n \nabla_\theta L_i(\theta^{\text{old}})\|^2 = \mathbb{E}\left[\mathbb{E}\left[\|\sum_{i=1}^n \nabla_\theta L_i(\theta^{\text{old}})\|^2 \mid p = n_p\right]\right]$$

$$= \mathbb{E}\left[\mathbb{E}\left[\|p\nabla_\theta U(\theta^{\text{old}}) + e^{\text{old}}\|^2 \mid p = n_p\right]\right]$$

$$= \mathbb{E}\left[\mathbb{E}\left[p^2\|\nabla_\theta U(\theta^{\text{old}})\|^2 + \|e^{\text{old}}\|^2 + 2p\langle\nabla_\theta U(\theta^{\text{old}}), e^{\text{old}}\rangle \mid p = n_p\right]\right]$$

$$= \mathbb{E}[p^2]\|\nabla_\theta U(\theta^{\text{old}})\|^2 + \mathbb{E}[\|e^{\text{old}}\|^2] + 0$$

$$\leq n^2\|\nabla_\theta U(\theta^{\text{old}})\|^2 + n\sigma^2.$$

Plugging in back to equation (8) we have

$$\mathbb{E}[U(\theta^{\text{new}})] \leq U(\theta^{\text{old}}) - \frac{\alpha P}{N}\|\nabla_\theta U(\theta^{\text{old}})\|^2 + \frac{\alpha^2 C}{2n^2}(n^2\|\nabla_\theta U(\theta^{\text{old}})\|^2 + n\sigma^2)$$

$$= U(\theta^{\text{old}}) - (\frac{\alpha P}{N} - \frac{\alpha^2 C}{2})\|\nabla_\theta U(\theta^{\text{old}})\|^2 + \frac{\alpha^2 C\sigma^2}{2n}.$$

Therefore, a sufficient descent such that

$$\mathbb{E}[U(\theta^{\text{new}})] \leq U(\theta^{\text{old}}) - c\alpha\|\nabla_\theta U(\theta^{\text{old}})\|^2,$$

for some $c \geq 0$ can be guaranteed by

$$(\frac{\alpha P}{N} - \frac{\alpha^2 C}{2})\|\nabla_\theta U(\theta^{\text{old}})\|^2 - \frac{\alpha^2 C\sigma^2}{2n} \geq c\alpha\|\nabla_\theta U(\theta^{\text{old}})\|^2$$

Rearrange

$$\frac{P}{N} \geq c + \frac{\alpha C\sigma^2}{2n\|\nabla_\theta U(\theta^{\text{old}})\|^2} + \frac{\alpha C}{2}$$

This completes our proof. □

## D   MORE IMPLEMENTATION DETAILS

We provide more details about victims' architectures and training. Our model architectures follow official implementations in Moyer et al. (2018); Zhao et al. (2019).

On Adult dataset, we use

- CFAIR:
    - Encoder: linear, representation $z \in \mathbb{R}^{60}$.
    - Discriminators: one hidden layer with width 50, using ReLU activation.
    - Classifier: linear.
    - Training: AdaDelta optimizer with learning rate 0.1, batchsize 512, epochs 50.
- CFAIR-EO:
    - Encoder: linear, representation $z \in \mathbb{R}^{60}$.
    - Discriminators: one hidden layer with width 50, using ReLU activation.

- **–** Classifier: linear.
- **–** Training: AdaDelta optimizer with learning rate $0.1$, batchsize $512$, epochs $50$.
- **ICVAE-US:**
  - **–** Encoder: one hidden layer with width 64, output representation $z \in \mathbb{R}^{30}$, using Tanh activation.
  - **–** Decoder: one hidden layer with width 64, using Tanh activation.
  - **–** Classifier: one hidden layer with width 64. using Tanh activation.
  - **–** Training: Adam optimizer with learning rate $0.001$, batchsize $512$, epochs $50$.
- **ICVAE-S:**
  - **–** Encoder: one hidden layer with width 64, output representation $z \in \mathbb{R}^{30}$, using Tanh activation.
  - **–** Decoder: one hidden layer with width 64, using Tanh activation.
  - **–** Classifier: one hidden layer with width 64. using Tanh activation.
  - **–** Training: Adam optimizer with learning rate $0.001$, batchsize $512$, epochs $50$.

On German dataset, we use

- **CFAIR:**
  - **–** Encoder: linear, representation $z \in \mathbb{R}^{60}$.
  - **–** Discriminators: one hidden layer with width 50, using ReLU activation.
  - **–** Classifier: linear.
  - **–** Training: AdaDelta optimizer with learning rate $0.05$, batchsize $64$, epochs $300$.
- **CFAIR-EO:**
  - **–** Encoder: linear, representation $z \in \mathbb{R}^{60}$.
  - **–** Discriminators: one hidden layer with width 50, using ReLU activation.
  - **–** Classifier: linear.
  - **–** Training: AdaDelta optimizer with learning rate $0.05$, batchsize $64$, epochs $300$.
- **ICVAE-US:**
  - **–** Encoder: one hidden layer with width 64, output representation $z \in \mathbb{R}^{30}$, using Tanh activation.
  - **–** Decoder: one hidden layer with width 64, using Tanh activation.
  - **–** Classifier: one hidden layer with width 64. using Tanh activation.
  - **–** Training: Adam optimizer with learning rate $0.001$, batchsize $64$, epochs $300$.
- **ICVAE-S:**
  - **–** Encoder: one hidden layer with width 64, output representation $z \in \mathbb{R}^{30}$, using Tanh activation.
  - **–** Decoder: one hidden layer with width 64, using Tanh activation.
  - **–** Classifier: one hidden layer with width 64. using Tanh activation.
  - **–** Training: Adam optimizer with learning rate $0.001$, batchsize $64$, epochs $300$.

During training, we followed GradMatch (Geiping et al., 2020) and did not shuffle training data after each epoch. For better comparison, victims were always initialized with random seed 1 to remove randomness during the pre-training procedure. In different replications, we selected different poisoning samples with different random seeds. Experiments that consist of 5 replications used seed 1 to 5 respectively.

# E MORE EXPERIMENT RESULTS

## E.1 PRACTICABILITY OF PERTURBING PRE-PROCESSED DATA

Many poisoning attacks on images classifiers perturb raw data (namely pixels, (Huang et al., 2020; Geiping et al., 2020)), in this work we perturb pre-processed data. However, this does not necessarily undermine the practical significance of our work. To see why attacking the pre-processed data is practical, we take a view from the scope of data security. Many FRL methods are applied in high-stakes domains, such as loan application or job market screening. Due to the privacy concern, Data anonymization has been used by more and more data providers to protect their data privacy and is often done as a part of the data pre-processing as discussed in Iyengar (2002); Ram Mohan Rao et al. (2018). In such cases, a malicious data provider can release a poisoned pre-processed (anonymized) dataset and launch the attack on victim models trained with it.

## E.2 FLD SCORE APPROXIMATES BCE LOSS

We evaluated how well the optimal BCE loss of logistic regression can be approximated by three FLD scores used in our experiments: FLD, sFLD, and EUC. To this end, we train each victim for 50 epochs and compute the empirical optimal BCE loss of a logistic regression to predict $a$ from representation $z$ after each epoch. Then we compare the trend of BCE loss versus FLD, sFLD, and EUC scores.

Figure 4 visualizes the negative value of FLD, sFLD, and EUC and associated BCE losses of four victims trained on Adult and German dataset with and without poisoning samples crafted by corresponding ENG attack after each epoch. In all cases, the three scores approximate how BCE loss changes very well.

## E.3 EFFECTS ON DP VIOLATIONS

Here we report increase of DP violations of victims attacked by three ENG-based attacks and AA-based baselines in Figure 5. As analyzed in Section 2, our attack successfully increased the DP violation significantly on most setting, clearly establishing their effectiveness.

## E.4 EFFECTS ON ACCURACY

Here we report change of accuracy of predicting $y$ from representation $z$ learned by victims attacked by three ENG-based attacks and AA-based baselines in Figure 6. In general, all attacks had relatively subtle influence on prediction accuracy, and our proposed ENG-based attacks often changed accuracy less. These results imply that poisoned representations are still of high utility, and it will be difficult for the victim trainer to identify the attack by checking the prediction performance alone.

## E.5 PERFORMANCE OF ELASTIC-NET PENALTY

Here we report more experimental results on evaluating the performance of elastic-net penalty under varying hyper-parameters. Figure 7 and 9 reports how elastic-net penalty affects the attack performance, corresponding $L_1$ and $L_2$ norms of learned perturbations are reported in Figure 10 and 11, respectively.

We further compare elastic-net penalty versus $L_1$ norm penalty for regulating the attack. Figure 8 shows corresponding decrease of BCE loss and Figure 12 shows corresponding $L_1$ and $L_2$ norms. Compared with elastic-net penalty, penalizing $L_1$ norm only usually resulted in larger $L_1$ and $L_2$ norms while the attack performance was hardly improved, especially when small- to intermediate-level coefficient(s) ($\lambda_1$ and $\lambda_2$) of the regularizer is used.

Given the advantages of utilizing elastic-net penalty, one may ask if it can be used in other attacks such as AA (Mehrabi et al., 2021) as well. Here we delve into the difficulty of such an application, highlighting the suitability of elastic-net penalty for our attack by nature.

According to Mehrabi et al. (2021), both RAA and NRAA constructed poisoning samples from a chosen anchor point by perturbing its $x$ randomly within a $\tau$-ball and flipping its $y$ or $a$. The

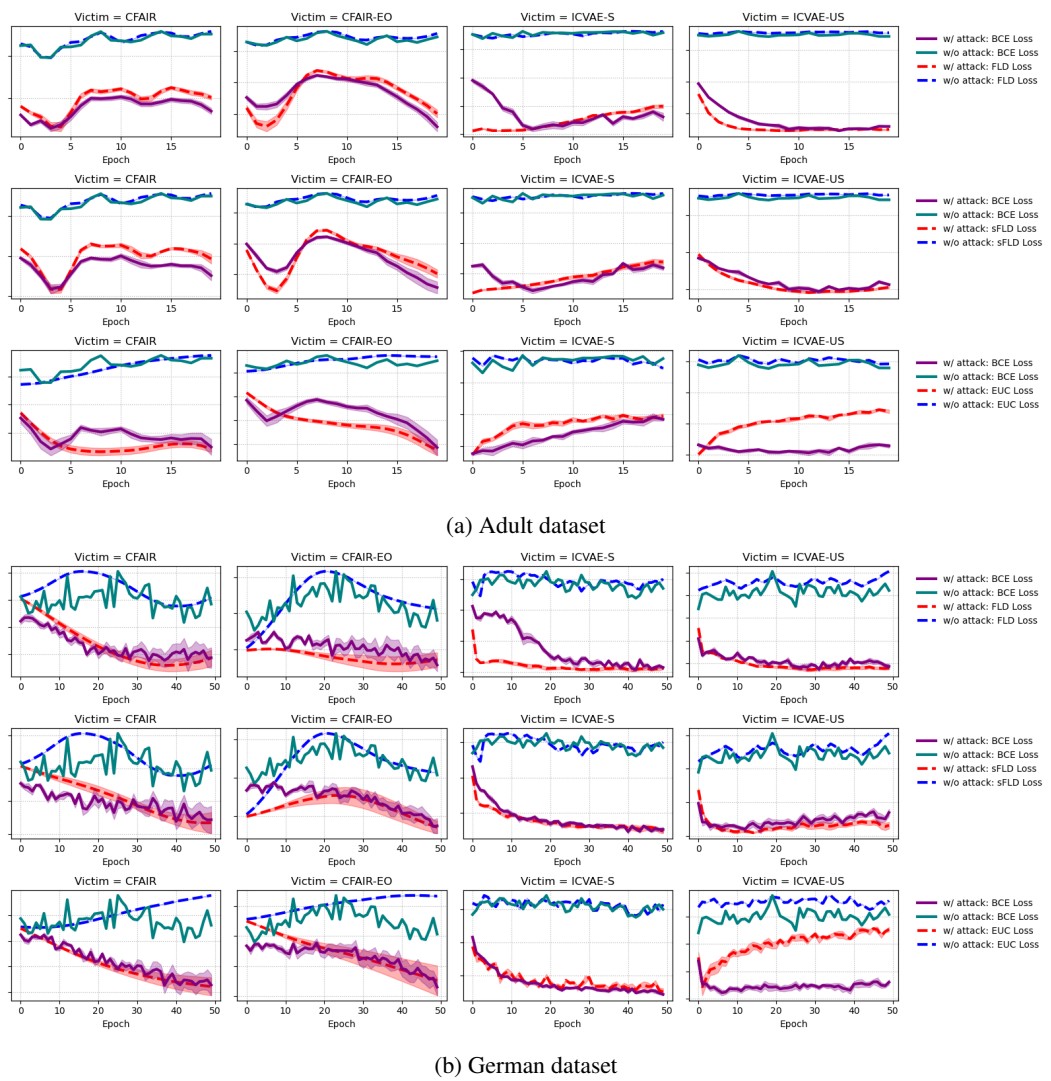

Figure 4: Changes of FLD, sFLD, and EUC loss (the negative score) and corresponding BCE loss. Results are averaged over 5 independent replications and bands show standard errors.

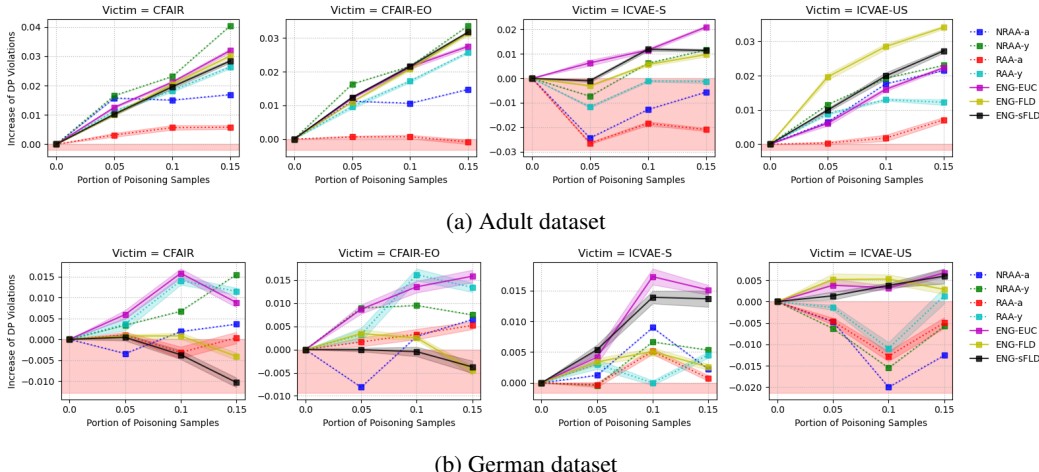

Figure 5: Increase of DP violations from different attackers using 5% - 15% training samples for poisoning, Results are averaged over 5 independent replications and bands show standard errors.

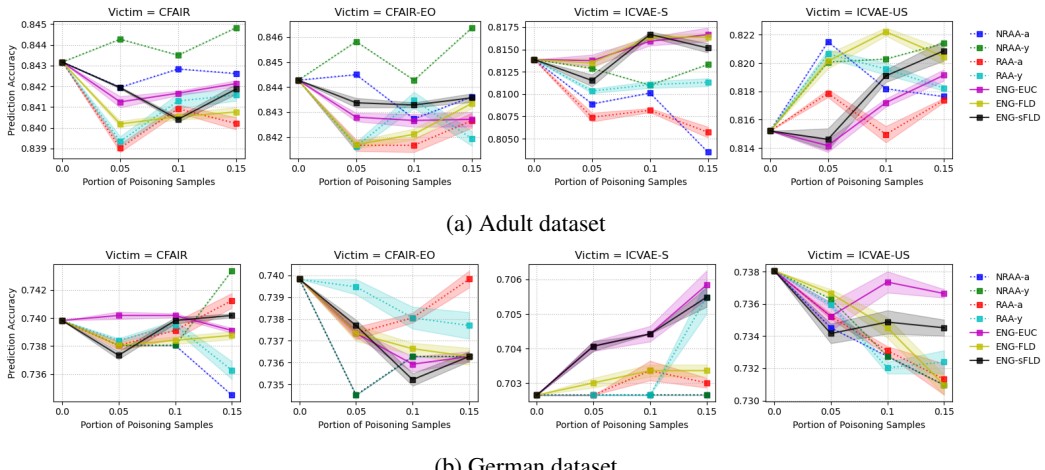

(a) Adult dataset

(b) German dataset

Figure 6: Accuracy of predicting $y$ from $\boldsymbol{z}$ when different attackers using 5% - 15% training samples for poisoning, Results are averaged over 5 independent replications and bands show standard errors.

main difference lies in how they choose these anchor samples: RAA draws them uniformly from the training data. NRAA computes the pairwise distance between all training data and counts how many *neighbors* and chooses the training samples that have the most neighbors with flipped $y$ or $a$ as anchor data. Here two samples are *neighboring* if their $y$ and $a$ are same and the difference in their $\boldsymbol{x}$ is smaller than a pre-specified threshold.

It is viable to define $\tau$-ball based on the elastic-net norm. Nonetheless, the use of $\tau$-ball will make NRAA harder to analyze in general. NRAA is thought stronger (which is empirically verified in both Mehrabi et al. (2021) and our experiment) because the chosen poisoning samples are likely to have higher influence as they have more neighbors. However, sampling from $\tau$-ball will make some poisoning samples have more neighbors while others have less. In fact, the authors of AA used $\tau = 0$ throughout their experiments and we adopted this setting. This difficulty remains unsolved when using elastic-net norm to induce the $\tau$-ball. In conclusion, ENG, as a perturbation-learning-based attack, has a better affinity to the elastic-net penalty than AA baselines.

### E.6 ATTACKING MULTI-CLASS SENSITIVE FEATURE

Here we show empirical results of attacking victims trained on Adult dataset using *race* as the sensitive feature. Due to data imbalance issue we keep *white* and *black* group as they are. All other sensitive groups are re-categorized as *others*. This results in a 3-way sensitive feature $a$ and we measure $I(\boldsymbol{z}, a)$ by the decrease of cross-entropy (CE) loss to predict $a$ from $\boldsymbol{z}$ with a linear Softmax classifier. Note that both the CE loss and the averaged one-vs-all BCE loss (see Appendix B for details) lower bounds $I(\boldsymbol{z}, a)$, but we report CE loss for the sake of better interpretability. Due to the lack of official implementation of CFAIR and CFAIR-EO on multi-class sensitive feature, we only attack ICVAE-US and ICVAE-S. As shown in Figure 13, ENG-based attack outperforms four AA baselines.

### E.7 DEFENSE AGAINST ENG ATTACKS

Here we present reducing batch size as a defense strategy against ENG attacks. We only defend against successful attacks and focus on Adult dataset where attacks used $\lambda_1 = 0.0025, \lambda_2 = 0.005$. We vary the batch size between 256 to 1024 and keep all other settings same as in Section 4. Resultant decrease of upper-level loss $-s$ and BCE loss are reported in Figure 14. Reducing batch size successfully helped weaken the performance of attacking 3 out of 4 victims in terms of $-s$.

Note that according to Theorem 3.4, it is conceptually viable to increasing learning rate $\alpha$ to defend. However, using a large learning rate in a well-trained victim model may have side effect of making it diverge. Because of this, we consider reducing batch size a better and more practical choice.

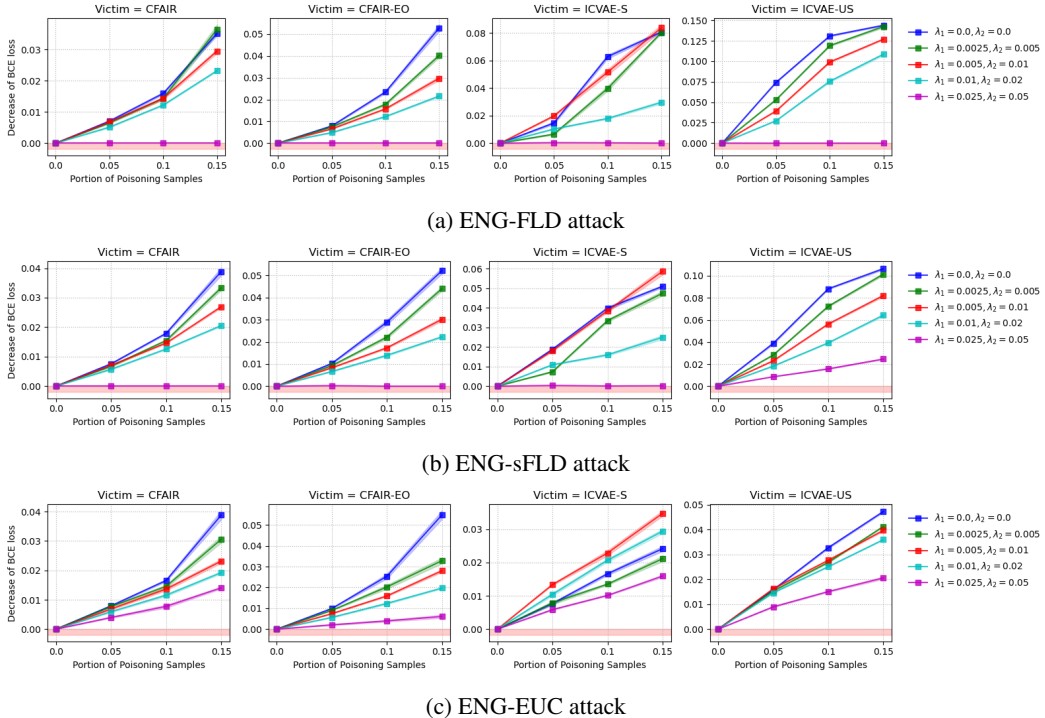

Figure 7: Decrease of BCE loss achieved by different ENG-based attacks with varying hyperparameters, victims are trained on Adult dataset. Results are averaged over 5 independent replications and bands show standard errors.

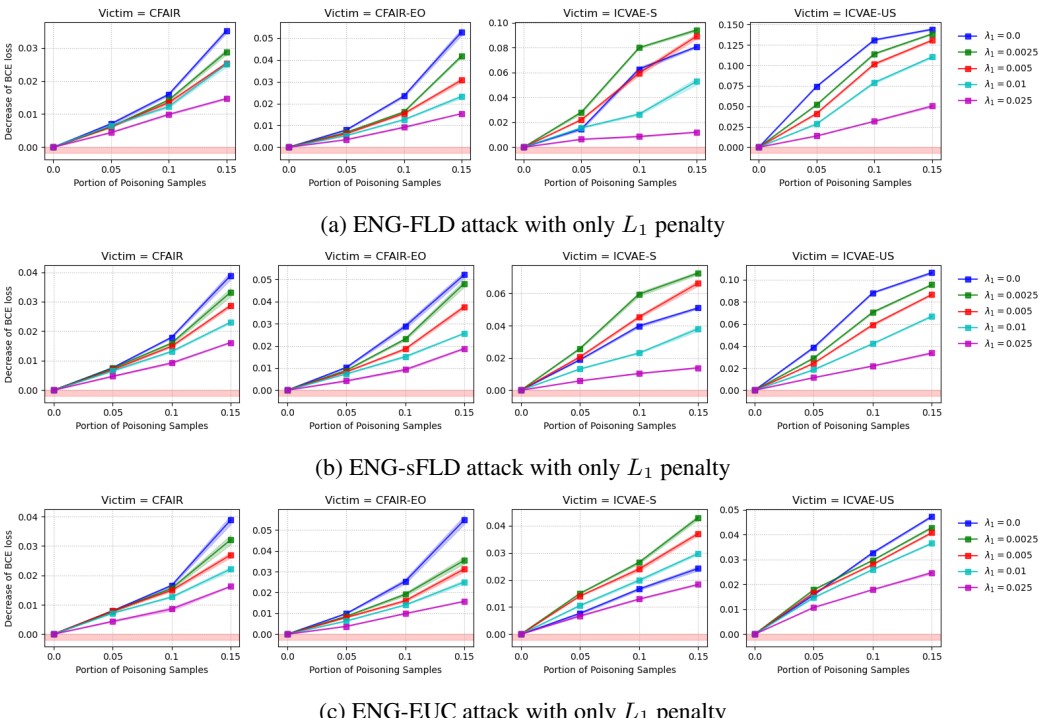

Figure 8: Decrease of BCE loss achieved by different ENG-based attacks with varying hyperparameters on $L_1$ penalty and no $L_2$ penalty, victims are trained on Adult dataset. Results are averaged over 5 independent replications and bands show standard errors.

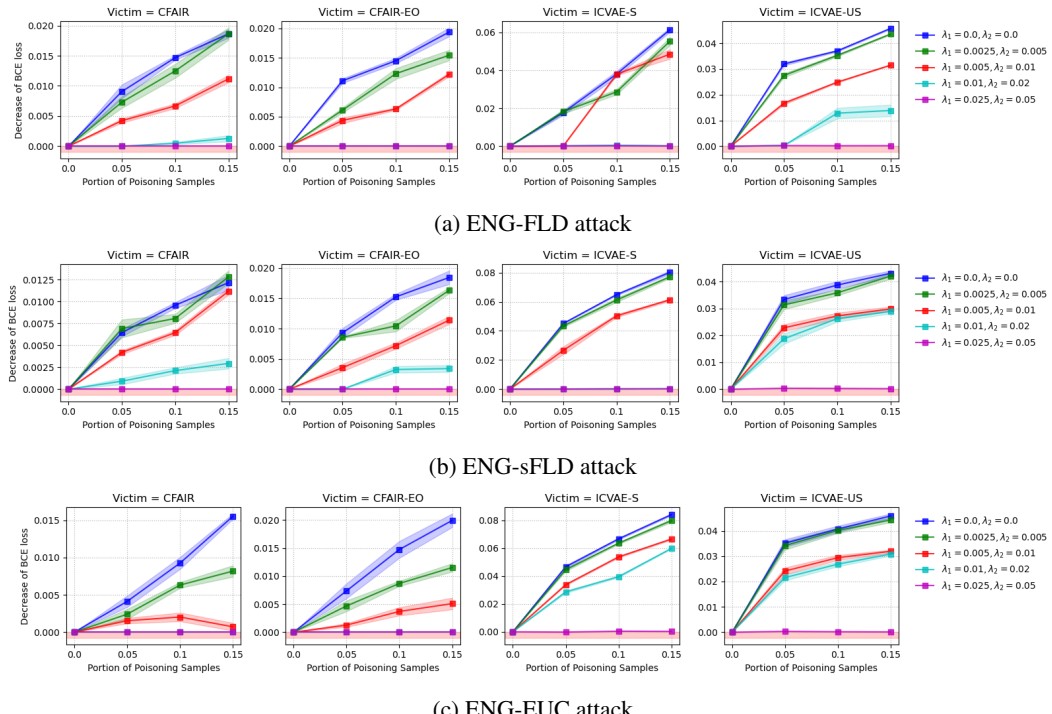

Figure 9: Decrease of BCE loss achieved by different ENG-based attacks with varying hyperparameters, victims are trained on German dataset. Results are averaged over 5 independent replications and bands show standard errors.

### E.8 MORE EXPERIMENT RESULTS ON COMPAS AND DRUG CONSUMPTION DATASETS

Here we present more experimental results of attacking four FRL methods trained on COMPAS and Drug Consumption datasets. We adopted the same setting for data preprocessing and training-testing splitting pipelines as presented in Section 4 on Adult and German datasets. In specific, we report decrease of BCE losses in Figure 15, increase of DP violations in Figure 16, and accuracy of predicting $y$ from $z$ in Figure 17. Again, our attacks succeeded on all cases, outperforming AA baselines to a large extent.

In terms of model architecture, we adopted the recommended architectures from Zhao et al. (2019) for CFAIR and CFAIR-EO on COMPAS dataset; and the default setting presented in Appendix D on Drug Consumption datasets due to the lack of official implementations. For ICVAE-US and ICVAE-S, we used the default setting from Appendix D on both datasets because of the lack of official implementations.

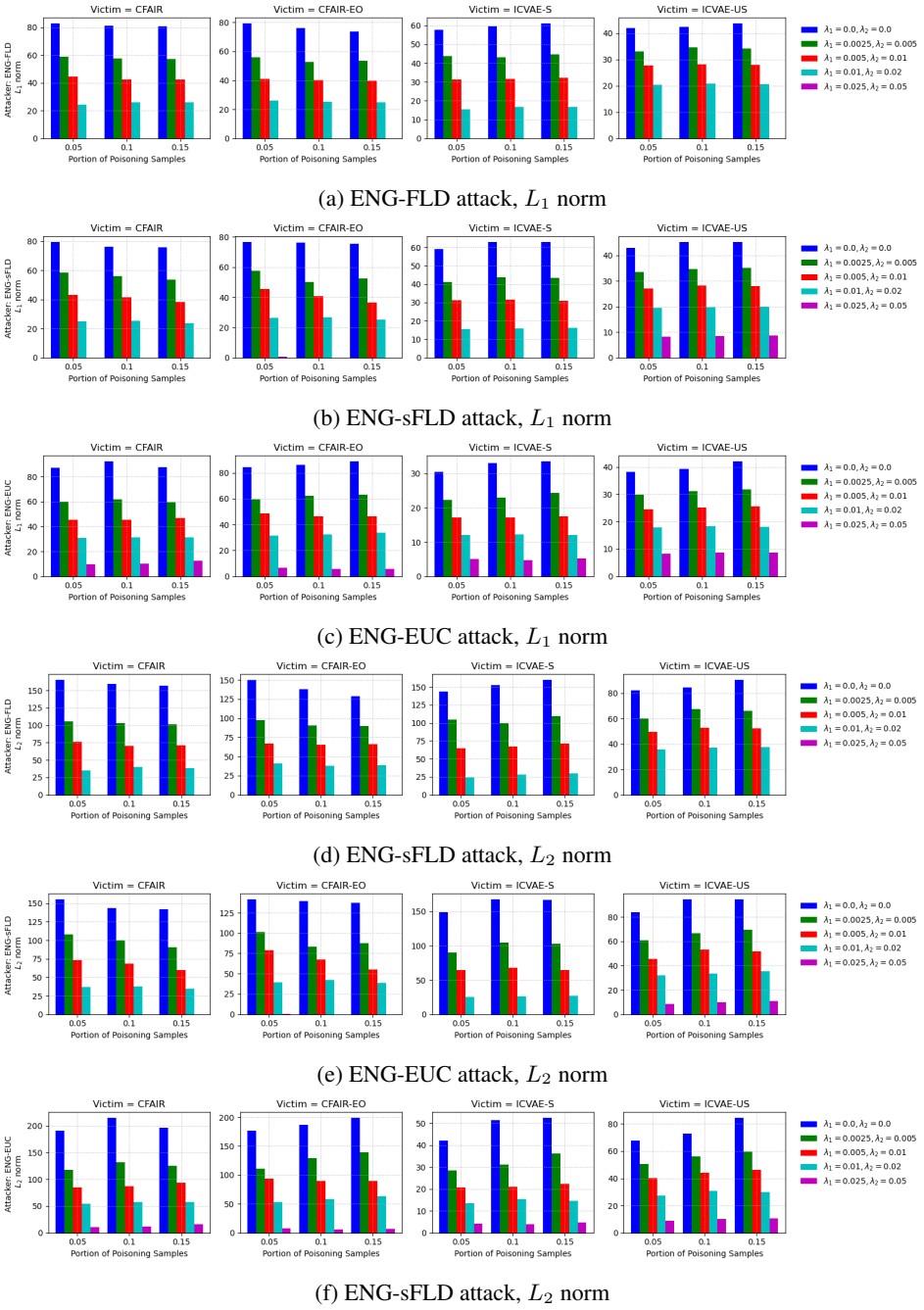

Figure 10: Averaged $L_1$ and $L_2$ norm of perturbations learned by different ENG-based attacks with varying hyper-parameters, victims are trained on Adult dataset.

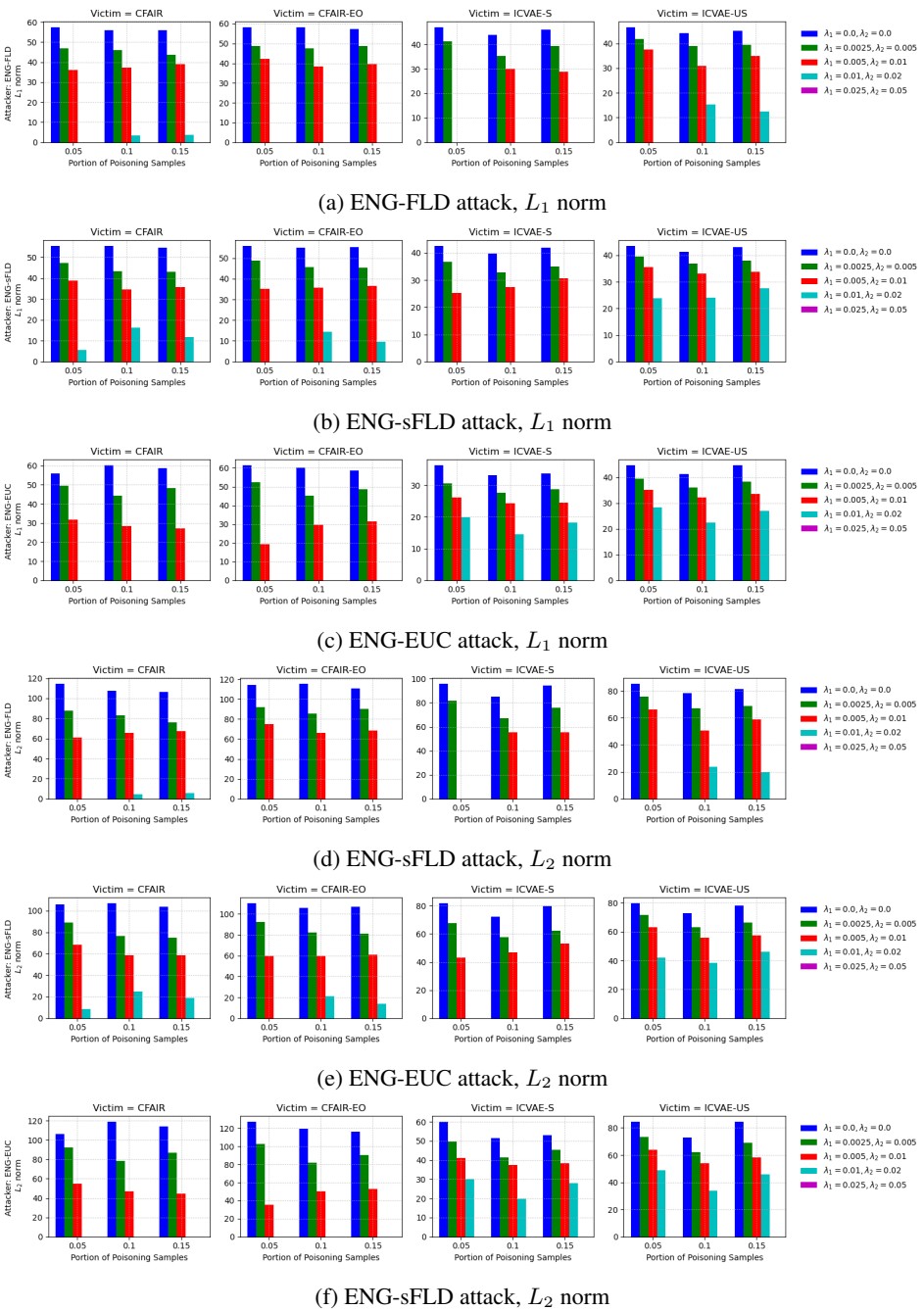

Figure 11: Averaged $L_1$ and $L_2$ norm of perturbations learned by different ENG-based attacks with varying hyper-parameters, victims are trained on German dataset.

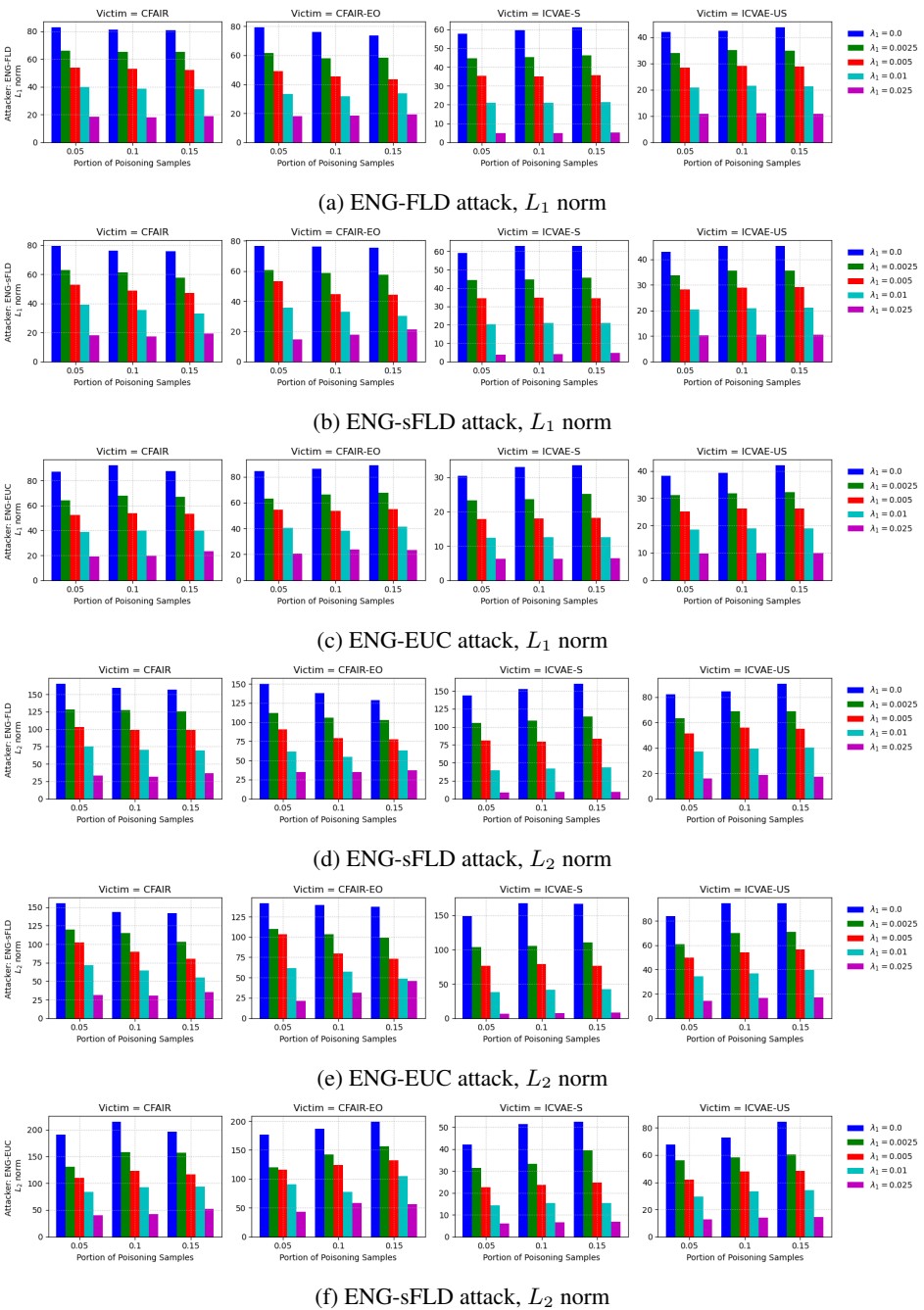

Figure 12: Averaged $L_1$ and $L_2$ norm of perturbations learned by different ENG-based attacks with varying hyper-parameters on $L_1$ penalty and no $L_2$ penalty, victims are trained on Adult dataset.

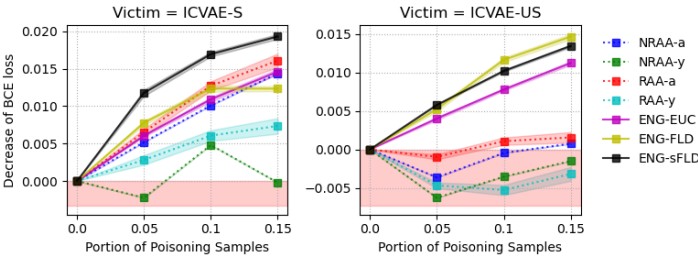

Figure 13: Decrease of CE loss from attacking victims trained on Adult dataset with sensitive feature *race*.

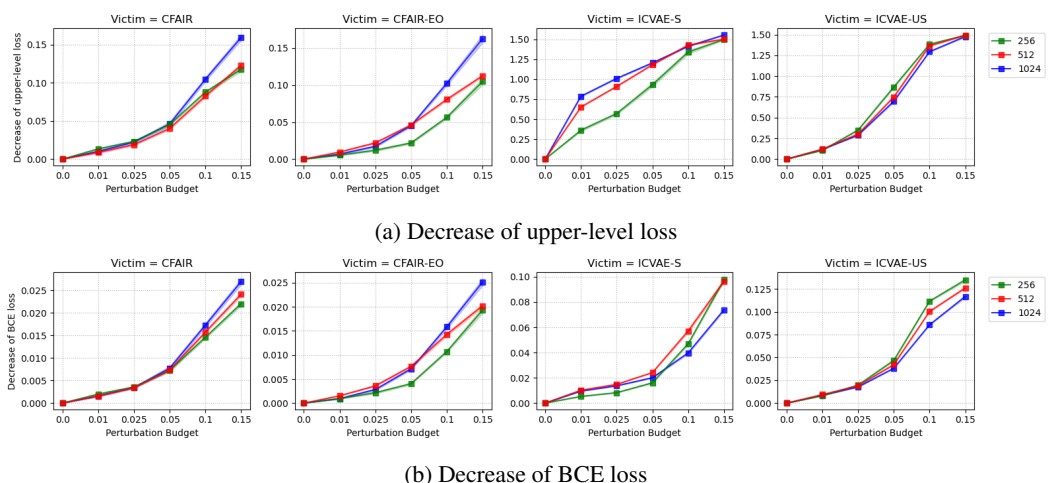

Figure 14: Effectiveness of reducing batch size as a defense against the proposed ENG-based attacks, different attackers using 1% - 15% training samples for poisoning. Results are averaged over 5 independent replications and bands show standard errors.

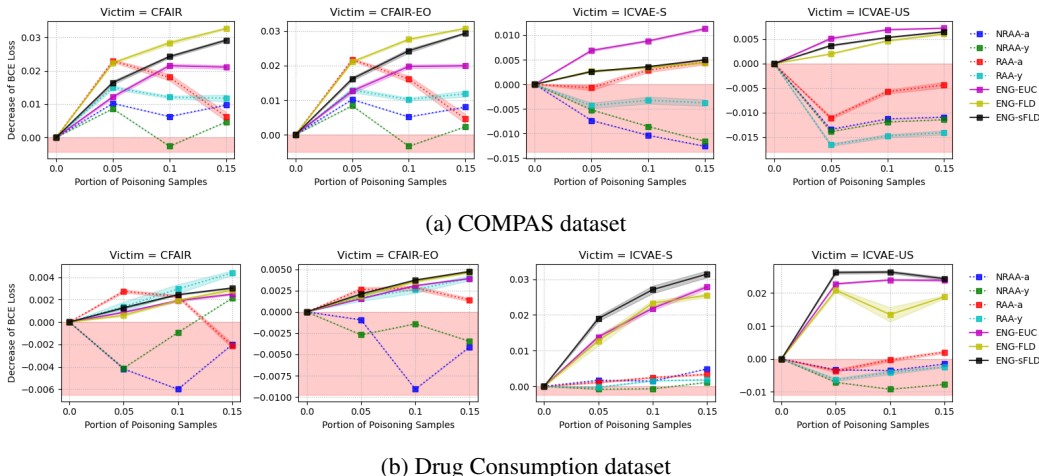

Figure 15: Decrease of BCE loss from different attackers using 5% - 15% training samples for poisoning, Results are averaged over 5 independent replications and bands show standard errors.

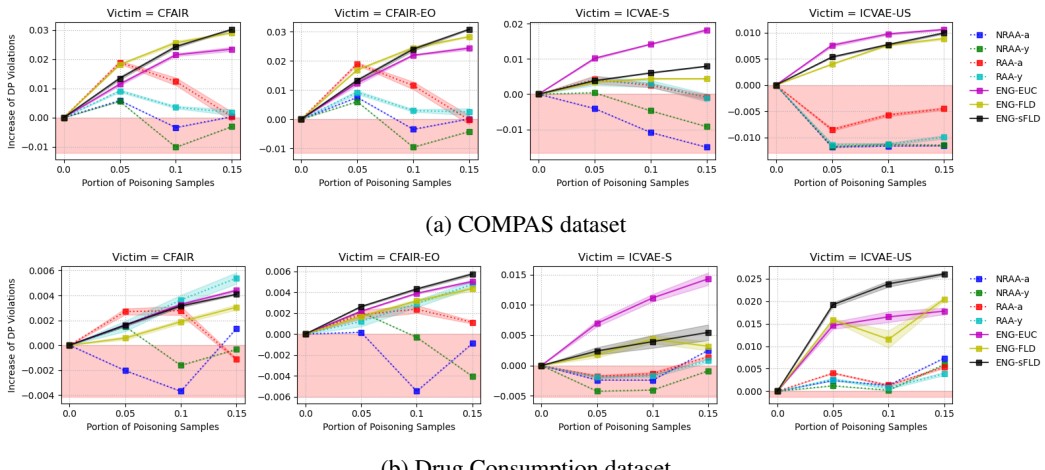

(a) COMPAS dataset

(b) Drug Consumption dataset

Figure 16: Increase of DP violations from different attackers using 5% - 15% training samples for poisoning, Results are averaged over 5 independent replications and bands show standard errors.

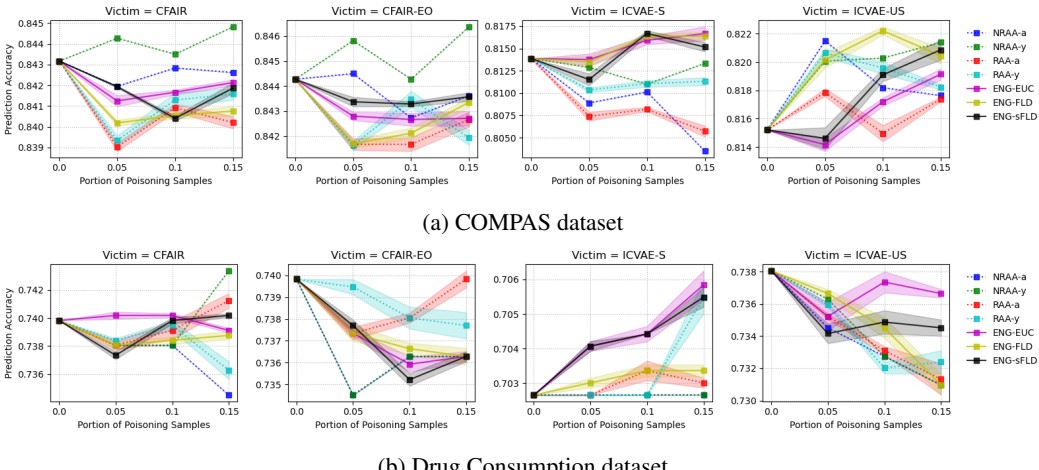

(a) COMPAS dataset

(b) Drug Consumption dataset

Figure 17: Accuracy of predicting $y$ from $z$ when different attackers using 5% - 15% training samples for poisoning, Results are averaged over 5 independent replications and bands show standard errors.

