# OpenReview forum: "Towards Poisoning Fair Representations"
_ICLR.cc/2024/Conference — ICLR 2024 poster_

### Official Review · Reviewer_LzqR · 2023-10-21

**Soundness:** 3 good
**Presentation:** 2 fair
**Contribution:** 3 good
**Rating:** 8
**Confidence:** 4

**Summary:**

This paper designs a new data poisoning attack tailored for fair representation learning. The key idea is to generate poisoning samples that maximizes the mutual information between the representation (of the poisoned sample) and the sensitive attribute. This can be solved by bi-level optimization, yet the outer level problem (estimating high-dimensional mutual information) is intractable. To solve this issue, the authors propose to use Fisher’s linear discriminant (FLD) score as a cheap proxy to MI, which has a closed-form solution. For inner level problem, it is solved by matching the gradients of a victim model and a clean model. Through the approximations made, the original objective is now fully tractable and can be learned by SGD. The effectiveness of the method is evaluated on two tabular datasets.

**Strengths:**

- **Significance**: The problem studied in this paper (data poisoning to attack the learned representation in fairly trained models) is interesting and important;
- **New information-theoretic viewpoint for poisoning attack**: The authors propose an information-theoretic objective for poisoning attack in fair machine learning. This framework, to the best of my knowledge, is new within the specific context considered and is very-well motivated. The reminder on the advantage of MI-based fairness as compared to conventional metric (e.g. DP) is also useful;
- **Cheap proxy to mutual information**: I also like the authors’ idea to use Fisher’s linear discriminant analysis (FLD) as a cheap-but-still-effective proxy to MI. I would also highly praise the authors’ efforts to mention the possibilities of other analytic proxies, as well as a discussing between FLD and these alternative methods.
- **Feature selection for fairness**: in addition to the main contribution, the authors also show how their developed attack can be further applied to identify robust features for fair classification. This is quite interesting, especially in that it offers a new perspective for understanding and interpreting the behavior of a fairly trained model. I personally think this part deserves a separate section and can be highlighted as a 2nd main contribution of the paper.

**Weaknesses:**

- **On the necessity of working at representation level**. After reading the paper, it is still unclear to me why do we need to consider I(Z; a) rather than I(Y(X); a). Here Y(X) is the prediction of the model. In fact, considering I(Y(X); a) has several benefits: first, its maximization still yields an unfair model; second, its estimation is much easier due to the low-dimensionality of Y(X) and a (for example, we can estimate it easily by the well-known KSG estimator [1] which typically works quite well in low-dimensional cases). Importantly, maximizing I(Y(X); a) also do not require an access to Y. Can the authors justify the reason behind considering I(Z; a) instead?

- **Issues in the discussion related to other MI proxies**: some comments in remark 2.1 do not seem completely sensible to me. For example, the author mention that other analytical proxies like (K)CCA and sliced mutual information suffer from differentiation difficulties. This may not be true in my view (for example, in CCA, you can first solve the optimal weights analytically, then substitute it back to the formula of CCA. The resultant formula has a structure very similar to eq2 in your paper). In addition, I think the authors miss the possibility of using non-parametric dependence measure e.g. distance correlation (dCorr) [2]. This is also fully analytical and may potentially be applicable in your scenario. Ultimately, I think the real advantage of your FLD-based method is it provably optimises a variational lower bound of MI, whereas other proxies (KCCA, slice MI, dCorr) may not. This seems to be a better justification of the use of your method.

- **Slightly limited evaluation**: most of the evaluations in this work are conducted on tabular data where the network size is small. Whether the method developed in this work will scale to larger networks e.g. those in computer vision and NLP remain unclear to me. However, given the nature of many existing literature in fairness research, which also only focus on tabular data, this should not be a main criticism for the paper.

[1] Estimating mutual information, Physical Review E, 2004

[2] Measuring and testing dependence by correlation of distances, Annals of Statistics, 2007

----------------------------------------------
Update after rebuttal:

The authors have addressed most of my concern above and I have updated my score to 8 accordingly.

**Questions:**

How does the proposed FLD-based method scale with the dimensionality of representation d? I am concerned about this since Gaussianity assumption typically violates in high-dimensional cases. Will the method still work well for e.g. d=128?

---

> ### Author Response · Authors · 2023-11-20
> **Rebuttal by Authors**
>
> We highly appreciate your effort and time spent reviewing our paper and thank you for your expertise and constructive comments. In the following, we address your comments and questions one by one.
>
> >**On the necessity of working at representation level.**
>
> Attacking mutual information at representation level aligns better with fair representation learning (FRL) and is more universal than attacking $I(Y(X), a)$. For example, a bank may want to obtain a representation for each user that can be used to determine their eligibility for **existing** and **upcoming** financial products such as credit cards without concerning about fairness again [1].
> Here each financial product is associated with a (unique) label, and determining the eligibility entails a classification task.
> In this case, there are two challenges for delivering a $Y(X)$-based attack.
> First, one has to determine and justify
> which classifier $Y(\cdot)$ to use and why consider the fairness of this specific classification task.
> Second, for any **upcoming** financial product, its label $Y$ does not exist and one cannot obtain classifier $Y(X)$ (one needs $Y$ to train such a classifier),
> let along attacking it.
> In contrast, a representation-level attack can overcome the two challenges in a single shot.
> As discussed in section 2.2,
> by maximizing $I(z, a)$, any $I(Y(X), a)$ **will be** maximized so long as the fairness concern exists.
> This implies launching attack on all labels $Y$ simultaneously, including the ones where classifies $Y(X)$ cannot be trained.
>
> >**Issues in the discussion related to other MI proxies.**
>
> We appreciate the detailed explanation and valuable feedback from the reviewer.
> We've revised the remark 2.2 in the paper as follows:
>
> >*Maximizing I(z,a) is a general framework to poison FRL and admits other proxies such as sliced mutual information (Chen et al., 2022), kernel canonical correlation analysis (Akaho, 2007), and non-parametric dependence measures (Szekely et al., 2007). In this work, we use FLD because of its conceptual simplicity, interpretability, and good empirical performance. As one may recognize, when $p(z | a = 1)$ and $p(z | a = 0)$ are Gaussian with equal variance, FLD is optimal (Hamsici \& Martinez, 2008) whose BCE loss attains the tight lower bound of $I(z, a)$ up to constant $H(a)$. In this case, our method provably optimize the lower bound of I(z, a) whereas other proxies
> whereas other proxies may not due to the lack of direct connections to mutual information. While the Gaussianity may not hold in general, FLD score as a measure of data separability is still valid (Fisher, 1936), and we verify its efficacy for our goal in Appendix D.2, where we show that FLD score is highly informative for the empirical minimal BCE loss of a logistic regression.*
>
> >**Slightly limited evaluation.**
>
> As pointed out by the reviewer, we focus on simple tabular data because many fair representation learning methods focus on these data [1, 2, 3, 4]. Moreover, previous work on attacking classical fair machine learning also only focused on tabular data [5, 6, 7], and we follow this convention.
>
> To better evaluate the effectiveness of our method, we added experiments on two more benchmark datasets COMPAS and Drug Consumption.
> Results have been updated in Appendix D.8.
> We reported decrease of BCE losses in Figure 15, increase of DP violations in Figure 16, and accuracy of predicting $y$ from $z$ in Figure 17.
> Again, on two new datasets our attacks successfully outperformed AA baselines to a large extent.
> Observing these good performance, we will study the vulnerability of bias mitigation methods for CV and NLP tasks as our future work.
>
> [1] Zhao et al. Conditional learning of fair representations, 2019.
>
> [2] Moyer et al. Invariant representations without adversarial training, 2018.
>
> [3] Creager et al. Flexibly Fair Representation Learning by Disentanglement, 2018.
>
> [4] Reddy et al. Benchmarking Bias Mitigation Algorithms in Representation Learning through Fairness Metrics, 2021.
>
> [5] Solans et al. Poisoning attacks on algorithmic fairness, 2020.
>
> [6] Mehrabi et al. Exacerbating algorithmic bias through fairness attacks, 2021.
>
> [7] Chang et al. On Adversarial Bias and the Robustness of Fair Machine Learning, 2020.

---

> > ### Author Response · Authors · 2023-11-20
> > **Rebuttal by Authors, Cont'd**
> >
> > >**How does the proposed FLD-based method scale with the dimensionality of representation d?**
> >
> > In our (revised) experiments, CFAIR and CFAIR-EO used dimension of representations ranging from 10 (COMPAS) and 60 (Adult and German).
> > We used model architectures and training setting adopted from the original paper [1] as we believe these choices were proper to learn fair representations.
> >
> > To better understand how our attack performs on larger $d$, we further tested how fair representations learned by CFAIR and CFAIR-EO can be attacked when their dimensions grows to 128. Experiments were conducted on the largest Adult dataset.
> > Due to time constraint, we just increased depths of all networks by 1 and all hidden sizes were set to 60 without tuning them. Other hyper-parameters such as training epochs, batch size, and learning rates were unchanged.
> > As shown in the table below, **our attacks succeeded in all settings and were able to deliver strong attacks.**
> >
> > Finally, we would like to mention that the choices of model architecture and training settings can have crucial influence on the performance with and without attack, so these results are just for illustration.
> >
> > | Victim | Attack | 5\% | 10\% | 15\% |
> > |------ | --------  | ----- | ----- | ----- |
> > |CFAIR | NRAA-a  |  0.0106 | 0.0186 | 0.0322 |
> > |      | NRAA-y  |  0.0238 | 0.0252 | 0.0303 |
> > |      | RAA-a   |  0.0066 | 0.0237 | 0.0338|
> > |      | RAA-y   |  0.0215 | 0.0175 | 0.0399|
> > |      | ENG-EUC |  0.004 | 0.0136 | 0.0272|
> > |      | ENG-FLD |  0.0131 | 0.0261 | 0.0574|
> > |      | ENG-sFLD|  0.0059 | 0.0118 | 0.0151|
> > |CFAIR-EO | NRAA-a   |  -0.0075 | 0.0029 | 0.0141|
> > |         | NRAA-y   |  0.0021 | 0.0096 | 0.0043|
> > |         | RAA-a    |  -0.0127 | -0.0131 | 0.0103|
> > |         | RAA-y    |  0.0083 | 0.0305 | 0.0114|
> > |         | ENG-EUC  |  0.0007 | 0.0093 | 0.0267|
> > |         | ENG-FLD  |  0.0061 | 0.0124 | 0.0244|
> > |         | ENG-sFLD |  0.0065 | 0.0113 | 0.0141|
> >
> > [1] Zhao et al. Conditional learning of fair representations, 2019.

---

> ### Comment · Reviewer_LzqR · 2023-11-21
> **Reply to author**
>
> I thank the authors for their effective rebuttal. Most of my concerns have been addressed. I therefore update the score to 8 in response to your efforts. The discussion on why to attack at a representation level is indeed insightful and instructive.
>
> My final suggestions are to (a) highlight the motivation for attacking at a representation level in the revised manuscript; (b) improve the presentation by e.g. avoiding inline figures and moving all figures to the top/bottom of a page; and (c) remind the reader the potential risk of violating the Gaussianity assumption in high-dimensional cases (and refer them to the ablation study). I would be very interested to see when will FLD begin to fail (and even if it fails for some d, it is still a good contribution).

---

> > ### Author Response · Authors · 2023-11-21
> >
> > We are glad to hear that our rebuttal is helpful for addressing your concerns and are grateful for your rescoring. We will update our paper to further improve its quality following your suggestions.

---

### Official Review · Reviewer_LCr6 · 2023-10-21

**Soundness:** 3 good
**Presentation:** 3 good
**Contribution:** 4 excellent
**Rating:** 6
**Confidence:** 3

**Summary:**

This work studies an interesting topic i.e. how to conduct data poisoning against fair representation learning tasks. Experiments are conducted on Adult and German datasets to demonstrate its effectiveness.

**Strengths:**

- Motivation is well-stated and interesting.
- Authors develop related  theoretical analysis on the needed number of poisoning samples is derived and shed light on defending against the attack.
- Personally I like the organization of Introduction section : ) It's clear and easy for reviewers to know the meaning of this work.

**Weaknesses:**

- Authors use their own defined vanilla metric, and lack related fairness-aware metrics like Equality odds (EO)
- Authors are encouraged to conduct more experiments on more datasets like COMPAS and Drug Comsumptionm, please kindly follow this AAAI paper which authors have cited: Exacerbating Algorithmic Bias through Fairness Attacks.
- Personally, I reckon authors are encouraged to conduct experiments on deeper NN (I think simple MLP is not that DEEP to be called "DNN"), though the datasets are relatively simple. I'm curious about these experiments to investigate ENG. Authors are encouraged to conduct more analysis on the further version of this work, which is good for community: )

**Questions:**

See above.

---

> ### Author Response · Authors · 2023-11-20
> **Rebuttal by Authors**
>
> We highly appreciate your effort and time spent reviewing our paper and thank you for your expertise and constructive comments. In the following, we address your comments and questions one by one.
>
> >**Authors use their own defined vanilla metric, and lack related fairness-aware metrics like Equality odds (EO).**
>
> In section 2.2 we analyzed how our attack can deteriorate the well-known fairness-aware metric demographic parity (DP).
> In experiments we verified that **our attack successfully exacerbated DP violations of predicting label $y$ from  representation $z$ learned by four victims on four datasets** (Adult, German, and new COMPAS, Drug Consumption). Due to page length these results were deferred to Appendix D.3 (Adult and German datasets) and Appendix D.8 (COMPAS and Drug Consumption datasets) respectively.
>
> >**Authors are encouraged to conduct more experiments on more datasets like COMPAS and Drug Consumption.**
>
> Thanks for your suggestions, we conducted experiments on COMPAS and Drug Consumption datasets and reported results in Appendix D.8 of our updated paper.
> We reported decrease of BCE losses in Figure 15, increase of DP violations in Figure 16, and accuracy of predicting $y$ from $z$ in Figure 17. Again, on two new datasets our attacks successfully outperformed AA baselines to a large extent.
>
>
>
> >**Personally, I reckon authors are encouraged to conduct experiments on deeper NN (I think simple MLP is not that DEEP to be called "DNN"), though the datasets are relatively simple.**
>
> To see how our attacks perform on deeper and larger NNs,
> we tested how fair representations learned by CFAIR and CFAIR-EO can be attacked when the NNs are larger. Experiments were conducted on the largest Adult dataset.
> We increased depths of encoder to 3 hidden layer and adversarial classifier to 5 hidden layer. We increased the dimension of representations to 128 and set all hidden sizes to 60. Due to time constraint, we did not tune the model architectures to obtain the best clean performance. Training epochs were increase from 50 to 100, and all other hyper-parameters such as batch size and learning rates were unchanged. As shown in the table below, **our attacks succeeded in all settings while three baselines suffered from several failures.**
>
> Finally, we would like to mention that the choices of model architecture and training settings can have crucial influence on the performance with and without attack, so these results are just for illustration. Moreover, motivated by the promising results, we plan to extend our attack to fair machine learning on large-scale image and text datasets.
>
> | Victim | Attack | 5\% | 10\% | 15\% |
> | ------ | ------------- | ----- | ----- | ----- |
>  CFAIR | NRAA-a  |  0.0047     | 0.0204 | -0.4015
> |  | NRAA-y  |  0.0163 | -0.278 | -0.3552
> |  | RAA-a   |  0.0086     | -0.0138 | -0.3312
> |  | RAA-y   |  0.011      | -0.0226 | -0.8761
> |  | ENG-EUC |  0.003      | 0.0058 | 0.01
> |  | ENG-FLD |  0.0019     | 0.0055 | 0.0105
> |  | ENG-sFLD|  0.0033     | 0.0052 | 0.0037
> | CFAIR-EO | NRAA-a  |  0.0171 | 0.0279 | -0.3513
> |  | NRAA-y  |  0.0194 | -0.3161 | -0.2261
> |  | RAA-a   |  -0.1668 | -0.2093 | -0.0896
> |  | RAA-y   |  -0.0585 | -0.5564 | -0.3719
> |  | ENG-EUC |  0.0021 | 0.007 | 0.0046
> |  | ENG-FLD |  0.0075 | 0.0128 | 0.0205
> |  | ENG-sFLD|  0.0073 | 0.0056 | 0.0213

---

> ### Comment · Reviewer_LCr6 · 2023-11-20
> **Reply to authors**
>
> Thank you for your good rebuttal. Question 1 is important to conduct this work since I see other reviewers also raise similar concerns, and authors conduct extra experiments on this concern. I think this work's contribution is improved after rebuttal so I rasie my rating for contribution evaluation.
> Hope other reviewers can retrospect the rebuttal content : )

---

> > ### Author Response · Authors · 2023-11-21
> >
> > We are glad to hear that our rebuttal is helpful for addressing your concerns and are grateful for your rescoring for the contribution evaluation. At the same time, please feel free to share with us any other suggestions if you find helpful for further improving the quality of our paper.

---

### Official Review · Reviewer_93wi · 2023-10-29

**Soundness:** 3 good
**Presentation:** 4 excellent
**Contribution:** 3 good
**Rating:** 8
**Confidence:** 3

**Summary:**

This study proposes a novel data poisoning attack against fair representation learning algorithms. Compared to previous attack methods against fair classification, this method proposes to craft the training dataset, in order to maximize the mutual information between the learned representation and sensitive raw features. This mutual information powered attack algorithm shows superior attack performances compared to anchor attacks against 4 different fair representation learning methods.

**Strengths:**

1/ This is the first research effort in organising data poisoning attacks against fair representation learning attacks. Different from fair classification problems, manipulating fair representation needs to control the statistical relation between high-dimensional embeddings  and raw feature inputs. This is challenging for directly extending previous fair learning poisoning methods. I'd appreciate the efforts poured towards this difficult problem.

2/ It is intuitive to increase the mutual information between the embeddings and sensitive raw features, in order to violate the fairness constraint of the victim embedding learning algorithm. However directly maximising mutual information of high-dimensional embeddings is very difficult. I know there are some differentiable approximation tool to MI, like MINE. But it is computationally costly and prone to the potential estimation gap. It is interesting to read the theoretical analysis and practices of using Fisher Linear Discriminator scores to bound MI. Apparently, optimising FLD scores is much easier and economic in computation.

3/ Inspired from Geiping et al's gradient matching work, this study propoes to matching the upper and lower bound of gradients instead of solving the bi-level poisoning problem. This smart optimization strategy enables an analytical solution to the proposed attack.

**Weaknesses:**

One of the problem of introducing elastic penalty is how to choose properly the two penalty parameters $\lambda_{1}$ and $\lambda_2$. Though it can be chosen empirically, it can be dataset-dependent. Would it make significantly difference if we simply choose the L1 norm penalty instead?

**Questions:**

Discussion over the choice of the two penalty parameters $\lambda_{1}$ and $\lambda_2$ in the elastic penalty.

---

> ### Author Response · Authors · 2023-11-20
> **Rebuttal by Authors**
>
> We highly appreciate your effort and time spent reviewing our paper and thank you for your expertise and constructive comments. In the following, we address your comments and questions one by one.
>
> >**One of the problem of introducing elastic penalty is how to choose properly the two penalty parameters...Would it make significantly difference if we simply choose the L1 norm penalty instead?**
>
> We chose the penalty parameters following the convention in previous work [1] and the choices were briefly selected from a relatively large range, therefore, we believe that the attack performance can benefit from a fine-grained search for better parameter values.
>
> Regarding the advantage of using elastic-net penalty than using $L_1$ norm penalty only,
> conceptually, elastic-net penalty is able to choose a group of correlated features whereas using $L_1$ or $L_2$ penalty only cannot [2].
> Empirically, to check the effect of using $L_1$ norm penalty only, we tried the same choices of $\lambda_1$ as in the main experiment
> and compare the corresponding attack performance and perturbation magnitude.
> Due to time constraint we only conducted experiments on Adult dataset and reported results in Appendix D.5 of the updated paper.
>
> In short, we note that **comparing with elastic-net penalty, $L_1$ norm penalty usually resulted in perturbations with larger $L_1$ and $L_2$ norms but the attack performance was hardly improved, especially when small- to intermediate- level penalty parameter values were used.**
>
> [1] Chen et al. Ead: elastic-net attacks to deep neural networks via adversarial examples, 2018.
>
> [2] Hui Zou and Trevor Hastie. Regularization and variable selection via the elastic net, 2005.

---

### Official Review · Reviewer_yTSY · 2023-11-01

**Soundness:** 3 good
**Presentation:** 3 good
**Contribution:** 2 fair
**Rating:** 5
**Confidence:** 3

**Summary:**

This paper studies data poisoning attacks against fair representation learning (FRL) on deep neural networks. To achieve the attack goal, the authors propose a new MI (mutual information) maximization paradigm. Besides, the experiments show that the proposed attack outperforms baselines by a large margin and raises an alert of the vulnerability of existing FRL methods.

**Strengths:**

This is a pioneering data poisoning attack on deep learning-based fair representation learning to degrade fairness while the existing fairness attacks are focusing on shallow-model classifiers.

The authors propose a new attack goal based on MI to amplify the difference between representations from different subgroups.
The authors derive the first theoretical minimal number of poisoning samples required by their attack, which is crucial for practical attacks.

**Weaknesses:**

The assumption of the threat model is strong. The proposed attack is under the assumption of a white-box threat model, where the attacker has full access to and control over the victim's trained model. This implies that the attack is primarily effective in scenarios where the victim has already trained a model and relies on the attacker's data for subsequent fine-tuning. Such a specific condition might limit the general applicability of the attack in diverse real-world scenarios.

Lack the reason why the method can attack FRL. The foundational principle of Fair Representation Learning (FRL) is to ensure fairness by removing sensitive features from the intermediate representation. The proposed attack, on the other hand, seeks to amplify the presence of sensitive information within these representations. The paper does not adequately elucidate why FRL techniques, designed to minimize sensitivity, are unable to counteract or mitigate the effects of the proposed attack. This leaves a gap in understanding the inherent vulnerabilities of FRL against the described attack strategy.

**Questions:**

Similar to weakness 1, could the authors clarify their threat model more clearly? Specifically, does the attacker need to know the victim’s trained model to generate poisoning data? If not, does the attacker only need to know the structure of the victim’s model and the model will be trained on the poisoning data from the attacker?

Similar to weakness 2, could the authors give more insight to explain why their attack cannot be mitigated by fair representation learning (FRL)?

Could the proposed attack be applied to deeper neural networks? The experiments of this paper are just on two-hidden layer CNNs.

---

> ### Author Response · Authors · 2023-11-20
> **Rebuttal by Authors**
>
> We highly appreciate your effort and time spent reviewing our paper and thank you for your expertise and constructive comments. In the following, we address your comments and questions one by one.
>
> >**Could the authors clarify their threat model more clearly?**
>
> As the first work towards conducting poisoning attack against fair representation learning (FRL) methods, we followed previous works on attacking classical fair machine learning such as [1, 2, 3] and assumed a strong attacker that has full knowledge to the victim model: including its architecture, training details, and parameters.
> This type of attack is known as white-box attack and has been widely used to benchmark the vulnerability (robustness) of a model [1, 2, 3, 4] and is necessary to develop a black-box attack in general (in the sense that an optimization-based attack is often conducted by attacking a collection of white-box surrogated victim models) [5, 6]. Therefore, we believe that this requirement does not undermine our contribution on understanding how vulnerable FRL methods are. Nonetheless, we agree with the reviewer that a black-box attack would be more practical and we will delve into developing stronger and more transferable attack towards FRL methods in our future work.
>
> [1] Solans et al. Poisoning attacks on algorithmic fairness, 2020.
>
> [2] Mehrabi et al. Exacerbating algorithmic bias through fairness attacks, 2021.
>
> [3] Chang et al. On Adversarial Bias and the Robustness of Fair Machine Learning, 2020.
>
> [4] Koh et al. Stronger data poisoning attacks break data sanitization defense, 2022.
>
> [5] Huang et al. MetaPoison: Practical General-purpose Clean-label Data Poisoning, 2021.
>
> [6] Geiping et al. Witches’ brew: Industrial scale data poisoning via gradient matching, 2021.
>
> >**Could the authors give more insight to explain why their attack cannot be mitigated by fair representation learning (FRL)?**
>
> Thanks for your question. We plan to explore this direction in our next step towards understanding the vulnerability of existing fair representation learning (FRL) method. Here we give two possible causes.
>
> First, fair machine learning methods often assume that the underlying data distribution is unchanged.
> When a distribution shift appears, many of these methods showed degraded performance [1, 2].
> We presume that FRL methods suffer from a similar vulnerability as well, i.e., their training is not robust enough against distribution shift.
> Consequently, a data poisoning attack, which injects carefully-crafted poisoning samples to the training data, can be seen as imposing an adversarial distribution shift to the empirical training distribution. This may deteriorate the fairness performance of FRL methods on the validation (targeted) data that follows the original distribution.
>
> Second, as mutual information is difficult to estimate and optimize, existing FRL methods also rely on approximate solutions that require training of some auxiliary models such as a sensitive feature predictor to conduct adversarial training or a deep neural network-based mutual information estimator.
> In practice, these auxiliary models may also encounter unstable training and result in sub-optimal fairness results, and the performance degradation can be exacerbated by the data poisoning attack.
>
>
> [1] An et al. Transferring Fairness under Distribution Shifts via Fair Consistency Regularization, 2022.
>
> [2] Jiang et al. Chasing Fairness Under Distribution Shift: A Model Weight Perturbation Approach, 2023.

---

> > ### Author Response · Authors · 2023-11-21
> > **Rebuttal by Authors, Cont'd**
> >
> > >**Could the proposed attack be applied to deeper neural networks?**
> >
> > To see how our attacks perform on deeper and larger NNs,
> > we tested how fair representations learned by CFAIR and CFAIR-EO can be attacked when the NNs are larger. Experiments were conducted on the largest Adult dataset.
> > We increased depths of encoder to 3 hidden layer and adversarial classifier to 5 hidden layer. We increased the dimension of representations to 128 and set all hidden sizes to 60. Due to time constraint, we did not tune the model architectures to obtain the best clean performance. Training epochs were increase from 50 to 100, and all other hyper-parameters such as batch size and learning rates were unchanged. As shown in the table below, **our attacks succeeded in all settings while three baselines suffered from several failures.**
> >
> > Finally, we would like to mention that the choices of model architecture and training settings can have crucial influence on the performance with and without attack, so these results are just for illustration. Moreover, motivated by the promising results, we plan to extend our attack to fair machine learning on large-scale image and text datasets.
> >
> > | Victim | Attack | 5\% | 10\% | 15\% |
> > | ------ | ------------- | ----- | ----- | ----- |
> >  CFAIR | NRAA-a  |  0.0047     | 0.0204 | -0.4015
> > |  | NRAA-y  |  0.0163 | -0.278 | -0.3552
> > |  | RAA-a   |  0.0086     | -0.0138 | -0.3312
> > |  | RAA-y   |  0.011      | -0.0226 | -0.8761
> > |  | ENG-EUC |  0.003      | 0.0058 | 0.01
> > |  | ENG-FLD |  0.0019     | 0.0055 | 0.0105
> > |  | ENG-sFLD|  0.0033     | 0.0052 | 0.0037
> > | CFAIR-EO | NRAA-a  |  0.0171 | 0.0279 | -0.3513
> > |  | NRAA-y  |  0.0194 | -0.3161 | -0.2261
> > |  | RAA-a   |  -0.1668 | -0.2093 | -0.0896
> > |  | RAA-y   |  -0.0585 | -0.5564 | -0.3719
> > |  | ENG-EUC |  0.0021 | 0.007 | 0.0046
> > |  | ENG-FLD |  0.0075 | 0.0128 | 0.0205
> > |  | ENG-sFLD|  0.0073 | 0.0056 | 0.0213

---

> > > ### Author Response · Authors · 2023-11-22
> > >
> > > Dear Reviewer yTSY,
> > >
> > > Thank you again for providing helpful feedback on our paper, and we hope our response has addressed your questions and concerns. Since the discussion phase is approaching an end, we are wondering if you have any additional questions or concerns. We always treasure the comments you might have. Last but not least, we would really appreciate your kind consideration in raising your score if you find our response satisfactory.
> > >
> > > Best Regards,
> > > Authors

---

### Official Review · Reviewer_Ddgu · 2023-11-06

**Soundness:** 2 fair
**Presentation:** 3 good
**Contribution:** 2 fair
**Rating:** 3
**Confidence:** 5

**Summary:**

The authors propose a poisoning attack strategy to compromise fair representation learning, aiming to increase the fairness gap for underprivileged groups. The attack relies on approximations to solve a bilevel optimization problem where the outer problem, which describes the attacker’s objective, aims to maximize the mutual information between the representation for the privileged and the underprivileged groups. Since the optimization of this objective is not tractable, the authors use Fisher’s Linear Discriminant (FLD) score as a proxy. Then, the whole bilevel optimization problem is approximated using a gradient matching strategy. The authors also provide some theoretical analysis on the poisoning ration required to compromise the target models.

**Strengths:**

+ Poisoning fair representations have received less attention in the research literature on data poisoning and preliminary works show that these attacks can have a significant impact on the fairness of the target algorithms. Exploring more scalable poisoning attack strategies capable of increasing the fairness gap for deep neural networks is timely and a topic of interest.

+ The authors strived to provide a theoretical analysis on the ratio of poisoning points required to compromise the target algorithms.

**Weaknesses:**

+ The paper lacks a clear threat model. For example, it is unclear whether the attacker’s objective is useful for compromising algorithms with and without mechanisms for mitigating the fairness gap. On the other side, it is unclear what is the attacker’s objective and the relation of the attack strategy with respect to the model’s performance. Other works in the research literature, like Chang et al. or Van et al. (“Poisoning Attacks in Fair Machine Learning”) have already considered the trade-off between targeting performance and the fairness gap.

+ In the end the attack proposed by the authors rely on the maximization of the FLD score in the outer optimization objective. This relies on strong assumptions on the distribution of the data and its representation and may not hold for many practical scenarios. In this sense, it is unclear why this strategy is better compared to other attacks already proposed in the research literature, like Solans et al., Mehrabi et al., Chang et al., or Van et al. (“Poisoning Attacks in Fair Machine Learning”), which have strong connections to this work. On the other side, there is not mention to “Subpopulation Poisoning Attacks” by Jagielski et al., which are also very relevant to this work and proposes more scalable alternatives for crafting poisoning attacks targeting fairness.

+ Given the existing works in the research literature, I believe it is to bold that the authors claim that “We propose the first data poisoning attack on FRL as outlined in Figure 1.” I think that the authors should clarify this and position better their paper and contributions with respect to other existing works.

+ The experimental evaluation is not convincing: In the experiments the authors just reported results evaluating the BCE loss but did not consider any of the existing metrics for measuring the fairness gap. On the other hand, there is not mention to how the attack affects the accuracy at all. Apart from that, it is necessary a more comprehensive comparison with other methods in the research literature, e.g., Solans et al., Chang et al. Van et al. (“Poisoning Attacks in Fair Machine Learning”), Jagileski et al. (“Subpopulation Poisoning Attacks”). For some of these attacks, the authors can use the same outer objective and use the same approximation for solving the bilevel optimization problem.

+ The authors used a gradient matching strategy for approximating the solution of the bilevel optimization problem. However, Jagielski et al. (“Subpopulation Poisoning Attacks”) shows that other strategies can be more efficient for this. I think this aspect requires further analysis.

+ The authors say: “Heuristics such as label flipping (Mehrabi et al., 2021) lack a direct connection to the attack goal, thereby having no success guarantee and often performing unsatisfactorily.” I think this is not true. Although more limited on the attacker’s capabilities, smart manipulation of the labels can lead to successful attacks. See for example “Subpopulation Poisoning Attacks” by Jagielski et al.

**Questions:**

+ Equation (2) relies on strong assumptions about the distribution (Gaussian distribution and continuous variables) of the different subpopulations. How is this a good proxy for approximating the original problem? How does this compare with existing attacks (as the ones mentioned before)?

+ Could the authors provide more details on the threat model (see comments above)?

+ How assumption 2 works in the current threat model for the attack? The authors say: “Before the attack, the victim is well trained.” What does this mean?

+ Also, for the theoretical analysis: Why do the authors think that assumption 3 is reasonable?

---

> ### Author Response · Authors · 2023-11-20
> **Rebuttal by Authors**
>
> We highly appreciate your effort and time spent reviewing our paper and thank you for your expertise and constructive comments. In the following, we address your comments and questions one by one.
>
> >**The paper lacks a clear threat model: it is unclear whether the attacker’s objective is useful for compromising algorithms with and without mechanisms for mitigating the fairness gap. On the other side, it is unclear what is the attacker’s objective and the relation of the attack strategy with respect to the model’s performance.**
>
> In this work we proposed a new objective to attack fair representation learning (FRL) models by maximizing the mutual information (MI) between high-dimensional representations $z$ learned by the victim and the sensitive attribute $a$.
> This attacking objective aligns well with the well-known MI-based fairness goals used in FRL as detailed in section 2.1.
> Moreover, this objective jeopardizes the well-known fairness metric demographic parity (DP) of any classifiers predicting label $y$ from representation $z$ as discussed in section 2.2, and is empirically verified in our experiments covering four FRL models trained on four well-studied benchmark datasets.
> We further added accuracy of predicting label $y$ from representation $z$, and found that **our attacks had minimal impact on the accuracy.**
>
> >**It is unclear why this strategy is better compared to other attacks already proposed in the research literature**
>
> Existing works, including the listed one, focused on attacking fairness in classification tasks, i.e., the fairness is directly defined and attacked based on scalar predictions.
> We instead seek to degrade the *fairness* of *high-dimensional* representations directly, which is a much more challenging task and requires a new formulation as discussed in section 1.
> Notably,
> FRL methods do not always need access to label $y$ during the training time, see ICVAE-US for an example. For such victims, all existing poisoning attack formulation fall short to apply while our formulation is able to handle them.
>
> >**There is not mention to “Subpopulation Poisoning Attacks” by Jagielski et al., which are also very relevant to this work and proposes more scalable alternatives for crafting poisoning attacks targeting fairness. & The authors used a gradient matching strategy for approximating the solution of the bilevel optimization problem. However, Jagielski et al. (“Subpopulation Poisoning Attacks”) shows that other strategies can be more efficient for this. I think this aspect requires further analysis.**
>
> Thanks for bringing to us this work.
> After reading through it, we agree that subpopulation attack is a direction that deserves more exploration in the future study on deriving stronger attack against FRL methods, and we have updated the paper to discuss this accordingly.
> At the same time, we found subpopulation attack less relevant to the current main focus of our paper.
> First, a subpopulation attack, which identifies and attacks the most vulnerable subgroups of data, fills the gap between the availability and targeted attack. While in this work, we focus on an availability-like attack, where the attacker seeks to degrade fairness guarantee for as many samples as possible (on the whole validation set).
> Second, as mentioned by the reviewer, subpopulation attack provides more scalable alternatives to craft poisoning samples,
> while the main focus of this paper is to formulate an attack goal towards fairness of high-dimensional representations and to propose a tractable proxy that can be used as an objective for poisoning samples crafting.

---

> > ### Comment · Reviewer_Ddgu · 2023-11-22
> >
> > Thank you very much for your comments and clarifications. I understand the contribution in terms of attacking FRL, but still the threat model deserves to be analyzed more carefully. I also believe that the experimental evaluation would require the comparison with other poisoning attacks in the research literature, apart from Mehrabi et al. I also see very relevant the comparison with the subpopulation poisoning attacks in Jagielski et al.

---

> ### Author Response · Authors · 2023-11-20
> **Rebuttal by Authors, Cont'd.**
>
> >**I think that the authors should clarify this and position better their paper and contributions with respect to other existing works.**
>
> The main contributions of our work lie in two-fold.
> First, we provide a new MI-based attack goal to degrade fairness of *high-dimensional* representations $z$. Specifically, we maximize $I(z, a)$ where $a$ is the sensitive feature.
> In contrast, previous attacks mostly focused on *scalar* predictions on $\hat{y} = h(z)$ and seeks to maximize $I(h(z), a)$ (loosely speaking) which is much easier to evaluate than $I(z, a)$ as $z$ is high-dimensional.
> As discussed in section 2.2, the benefit of using this new MI-based attack goal is that by maximizing $I(z, a)$, for any classifier $h$, $I(h(z), a)$ is free to grow (so DP violation increases [1]) so long as the fairness concern exists. Notably, this condition holds even if ground truth $y$ is unobserved so that previous attack goals cannot be defined.
>
> Second, we provide a theoretical analysis on the required poisoning samples to launch a gradient-matching based attack.
> This analysis also sheds light on defending against the proposed attack method and can be of independent interest to other gradient-matching based attackers.
>
> To avoid possible confusion, we've updated this statement as "*We propose the first data poisoning attack that directly targets on fairness of high-dimensional representations as shown in Figure 1.*"
>
> [1] Zemel et al. Learning Fair Representations, 2013.
>
> >**The experimental evaluation is not convincing: In the experiments the authors just reported results evaluating the BCE loss but did not consider any of the existing metrics for measuring the fairness gap.**
>
> We reported violation of demographic parity (DP), one of the most widely-used fairness notion, from the four victim models trained on Adult and German datasets in Appendix D.3.
> In the revised paper we further added corresponding results on COMPAS and Drug Consumption datasets in Appendix D.8 of updated paper.
> **In all cases, our attacks successfully increased DP violations, clearly showing its effectiveness.**
>
> >**For some of these attacks, the authors can use the same outer objective and use the same approximation for solving the bilevel optimization problem.**
>
> As analyzed in the responses above, we believe that previous outer objectives may not be good choices for our purpose and they indeed fail to apply in certain cases.
> For example, in [1] the outer objective is a weighted average of classification accuracy and relaxed group fairness gap.
> However, ICVAE-US does not involve any classification in its training, therefore, these two terms are undefined.
>
> [1] Van et al. Poisoning Attacks on Fair Machine Learning, 2021.
>
> >**The authors say: “Heuristics such as label flipping (Mehrabi et al., 2021) lack a direct connection to the attack goal, thereby having no success guarantee and often performing unsatisfactorily.” I think this is not true. Although more limited on the attacker’s capabilities, smart manipulation of the labels can lead to successful attacks.**
>
> Thanks for your thought. We meant to express that label flipping-based attacks do not directly optimize the attack goal, and the goodness of these manipulations can be hard to quantify. As a result, one may not provide theoretical analysis on the minimal number of poisoning samples as we did for the gradient-matching based attack. However, we agree that smart label flipping attack can lead to successful attacks in practice, and we have updated this statement as "*Heuristics such as label flipping (Mehrabi et al., 2021) do not directly optimize the attack goal, thereby often requiring great effort to design good manipulations to success.*"

---

> > ### Comment · Reviewer_Ddgu · 2023-11-22
> >
> > Thanks for your comments. As mentioned before, I think that the paper needs to compare with other methods in the state of the art, taking into account different fairness metrics. I think that measuring the BCE loss, as reported in the main results of the paper is not the best to assess the quality of the attack in this type of scenarios. On the other hand, the results shown in the appendix (Figure 5) for DP, do not show a clear advantage of the proposed method with respect to the other competing method. In some cases, as for example, for the German datasets, we can see that the DP score decreases when increasing the number of poisoning points. I believe this deserves a more detailed analysis and these experiments should be in the main part of the paper, not in the appendix.

---

> ### Author Response · Authors · 2023-11-20
> **Rebuttal by Authors, Cont'd (2)**
>
> >**Equation (2) relies on strong assumptions about the distribution (Gaussian distribution and continuous variables) of the different subpopulations. How is this a good proxy for approximating the original problem? How does this compare with existing attacks (as the ones mentioned before)?**
>
> Fair representations are often real-valued [1, 2, 3, 4] and thereof we believe that the "continuous variable" assumption naturally holds in general fair representation learning scenarios.
>
> In terms of the Gaussian assumption, as discussed in remark 2.2, when this assumption holds, Eq (2) is not only a proxy, but also an optimal solution. When this assumption does not hold, using FLD (Eq (2)) as a data separability measure is still valid [5] and we believe that it is straightforward to use data separability as a measure of how difficult it is to classify them and as a proxy for the optimal BCE loss one can achieve on this classification task (which also depicts the classification difficulty). Notably, this BCE loss is always a lower bound of the mutual information in the original problem. In comparison, to our best knowledge, there is no analysis on how existing attack goals (as the ones mentioned before) relate to the mutual information between high-dimensional representations $z$ and sensitive feature $a$.
>
> Empirically, Figure 4 in Appendix D.2 verified that FLD score acted pretty well in approximating the optimal BCE loss to classify different classes of fair representations learned by four different fair representation learning methods. In short, in this work, we use FLD because of its conceptual simplicity, interpretability, and good empirical performance.
>
> [1] Zhao et al. Conditional learning of fair representations, 2019.
>
> [2] Moyer et al. Invariant representations without adversarial training, 2018.
>
> [3] Creager et al. Flexibly Fair Representation Learning by Disentanglement, 2018.
>
> [4] Reddy et al. Benchmarking Bias Mitigation Algorithms in Representation Learning through Fairness Metrics, 2021.
>
> [5] Fisher. The use of multiple measurements in taxonomic problems, 1936.
>
> >**How assumption 2 works in the current threat model for the attack? The authors say: “Before the attack, the victim is well trained.” What does this mean?**
>
> A gradient matching-based attack relies on a *trained* victim model [1].
> To simplify the theoretical analysis, assumption 2 states that this trained model is converged to a local stationary point of the lower-level objective before being attacked.
> This assumption is reasonable because the lower-level objective is the one that the model is trained on.
> Moreover, it allows us to bound the influence of clean gradient from clean training data, and to derive a lower bound for the minimal number of poisoning samples. This bound is valid in the sense that when the assumption is violated, typically a larger amount of poisoning samples is needed.
>
> The rational behind is as follows.
> When training the victim on poisoned data, each mini-batch consists of two types of samples: poisoning samples will push the victim model to optimize the upper-level objective (attack goal), whereas clean sample will counteract with this goal and instead force the victim to optimize the lower-level objective. By assuming that the victim has converged with respect to the lower-level objective, how clean samples counteracts with poisoning samples will be minimized: they only add randomness to the poisoning gradients. This allows us to derive the bound in a similar way to stochastic gradient descent.
>
> [1] Geiping et al. Witches’ brew: Industrial scale data poisoning via gradient matching, 2020.
>
> >**Also, for the theoretical analysis: Why do the authors think that assumption 3 is reasonable?**
>
> Similar to assumption 2,
> assumption 3 allows us to simplify the derivation of the lower bound.
> This assumption assumes that all poisoning samples in hand are well-crafted.
> This assumption can be achieved by conducing the gradient matching on poisoning samples one by one.
> As gradient matching is a standard optimization problem, modern optimizers can solved it readily.
> In addition, assumption 3 does not require all poisoning samples to match the upper-level gradient exactly.
> Instead, each of them only needs to form an unbiased estimation.
> Notably, when the assumption is violated, typically a larger amount of poisoning samples is needed, and the derived lower bound is still valid.

---

> > ### Author Response · Authors · 2023-11-22
> >
> > Dear Reviewer Ddgu,
> >
> > Thank you again for providing helpful feedback on our paper, and we hope our response has addressed your questions and concerns. Since the discussion phase is approaching an end, we are wondering if you have any additional questions or concerns. We always treasure the comments you might have. Last but not least, we would really appreciate your kind consideration in raising your score if you find our response satisfactory.
> >
> > Best Regards,
> > Authors

---

> > ### Comment · Reviewer_Ddgu · 2023-11-22
> >
> > Thanks for your comments. Gradient matching attacks are typically used for targeted attacks but for more indiscriminate attacks with a higher amount of poison, assumption 2 can be too strong. This should be discussed further. Actually, Jagielski et al. ("Subpopulation data poisoning attacks") make a good analysis of different approximations to this type of optimization problems. I believe this aspect deserves more attention.
> >
> > Apart from this, I see that the first approximation to solve the problem with the BCE loss is reasonable. However, the justification of the approximation in (2) is unclear and, again, it relies on some assumptions that may be reasonable or not. Again, I think this aspect deserves more attention.

---

> > ### Comment · Reviewer_Ddgu · 2023-11-22
> >
> > After all these comments, I would like to thank the authors for all their effort in addressing my comments and the comments from the other reviewers. I do believe that the research direction of the paper is very interesting, but I also believe that the paper can be improved in different ways: 1) justifying better the threat model and the assumptions made and, perhaps, relaxing some of them; 2) providing a more comprehensive and convincing experimental evaluation, comparing with other methods in the state of the art and paying more attention to the typical metrics used for fair ML.
> > Although I appreciate the effort, I am keeping my score.

---

> > > ### Author Response · Authors · 2023-11-23
> > > **Thanks for your feedback, and response to your follow-up concerns**
> > >
> > > We are happy to know that our response is helpful for clarifying some other questions such as the contribution of directly attacking fair representations. Please see below our answers to your follow-up feedbacks one by one.
> > >
> > > >**Threat model deserves to be analyzed more carefully. Reasonability of approximation in (2).**
> > >
> > > The main idea of approximation in (2) is that the optimal BCE loss is not optimizable in general.
> > > Instead, we approximately minimize it by making data more separable.
> > > As a measure of data separability [1], FLD score serves as a proxy and a tractable objective for direct representation optimization.
> > >
> > >  **We use FLD because of its conceptual simplicity, interpretability, and good empirical performance.**
> > > Theoretically, FLD is **optimal** when the Gaussianity assumption holds [2] whose BCE loss attains the tight lower bound of $I(z, a)$ up to constant $H(a)$.
> > >     In this case, **our method provably optimize the lower bound of $I(z, a)$ whereas other threat models lack such guarantees due to the lack of direct connection to mutual information.**
> > >     Empirically, while the Gaussianity may not hold in general, FLD score as a measure of data separability is still valid [1] and **we verify its efficacy for our goal in Appendix D.2**, where we show that FLD score is highly informative for the empirical minimal BCE loss of a logistic regression.
> > >
> > > These aspects have been detailed in Section 2.3, and the condition of Gaussianity is discussed in remark 2.2.
> > >
> > > [1] Fisher. The Use of Multiple Measurements in Taxonomic Problems. 1936.
> > >
> > > [2] Hamsici and Martinez. Bayes optimality in linear discriminant analysis, 2008.
> > >
> > > >**Comparison with smart label-flipping attack for subpopulation attack.**
> > >
> > > We agreed that subpopulation attack [1] is a direction that deserves more exploration in the future work of deriving stronger attack against FRL methods. However, because of the **significant difference in attacker's capacities** as admitted by the reviewer, we believe that it is more reasonable to defer such a comparison to the future work.
> > >
> > > In specific, as recognized in the literature [2, 3, 4, 5], label-flipping attacks are usually considered having higher capacity to control the labeling process (and thereof less favored) than a clean-label attack such as ours.
> > > Moreover, subpopulation attack requires access to **an auxiliary dataset where any samples can be freely used as poisoning samples**. Notably, this auxiliary dataset can be **as large as 50\% of the total training data** [1].
> > > In contrast, our attack can only craft the **non-sensitive features of the certain portion of poisoning samples (at most 15\% in experiment)** that are **randomly chosen** before the victim training.
> > > In conclusion, we believe that **these significant differences will make a direct comparison unfair.**
> > >
> > > [1] Jagielski et al. Subpopulation Data Poisoning Attacks, 2021.
> > >
> > > [2] Shafahi et al. Poison frogs! targeted clean-label poisoning attacks on neural networks, 2018.
> > >
> > > [3] Geiping et al. Witches’ Brew: Industrial Scale Data Poisoning via Gradient Matching, 2021.
> > >
> > > [4] Paudice et al. Label sanitization against label flipping poisoning attacks, 2018.
> > >
> > > >**Assumption 2 can be too strong with a higher amount of poison**
> > >
> > > Thanks for your thought.
> > > Assumption 2 states that the victim trained with its own loss is converged to a stationary point.
> > > Its reasonability comes from two parts.
> > > (1) There exists such stationary points.
> > > this condition is weaker than assuming the lower-level optimization in Eq (1) (training the victim on poisoned data) has optimal solutions. Note that the set of optimal solutions form the feasible set of the upper-level optimization in Eq (1).
> > > If no stationary point exists, then Eq (1) (is widely used to formulated data poisoning attacks) will be ill-defined.
> > > Therefore, we reasonably expect this condition to hold.
> > > (2) The victim is converged to a local optimum.
> > > Note that this assumption is reasonable under regular conditions if the training of the victim can be run sufficiently many steps.
> > > In practice, we agree that with a higher amount of poison, this assumption can be harder to achieve.
> > > Nonetheless, We would like to clarify that **when the assumption is violated, the influence of clean samples will be underestimated and typically
> > > more poisoning samples are needed than the one given by the derivation.** In this sense, the bound is still valid but loose.
> > > Moreover, **we expect that the practical significance, i.e., how one can defend against the gradient matching-based attack by maximizing our derived lower bound, still holds.**
> > > Finally, we appreciate the reviewer's thought on this, we will leave a remark to discuss this in the revised manuscript.

---

> > > > ### Author Response · Authors · 2023-11-23
> > > > **Thanks for your feedback, and response to your follow-up concerns, Cont' d**
> > > >
> > > > >**The results shown in the appendix (Figure 5) for DP, do not show a clear advantage of the proposed method with respect to the other competing method. In some cases, as for example, for the German datasets, we can see that the DP score decreases when increasing the number of poisoning points.**
> > > >
> > > > We respectively disagree that the advantage of our attack is unclear.
> > > > Firstly, **only our attack succeeded in attacking DP metric of all four victims trained on four datasets** whereas baselines encountered severe failures.
> > > > Moreover, we would like to highlight that our attack did not target on maximizing DP violations for the classification task where DP violation is evaluated.
> > > > Instead, as analyzed in Section 2, conceptually, it simultaneously increases DP violations on **all possible classification tasks wherever fairness concern exists, including the ones that label are not observed and DP violation cannot be directly computed**.
> > > >
> > > > Regarding the decrease in the magnitude of DP violation exacerbation (but DP violation was still exacerbated)
> > > > when using more poisoning sample on German dataset, we presume that this is caused by the stability issue in optimization.
> > > > We note that such instability was only found on small datasets such as German (1k) and Drug Consumption (\~1.8k) datasets and was much mild, if not disappeared, on larger Adult (\~48k) and COMPAS (\~6k) datasets.
> > > > Moreover, baseline methods, if not failed, often suffered from severer degradtion of attack performance on these datasets.
> > > > We will add a description on this aspect and we will include DP violation results to the main body in the revised manuscript following the suggestion.
> > > >
> > > > >**Experimental evaluation can be more comprehensive and convincing**.
> > > >
> > > > We believe that main contribution of this work has been well-supported by our experiments.
> > > > Namely, we have the following main contributions:
> > > >
> > > > - We derived a new attack goal directly against fairness of high-dimensional representations based on a mutual information maximization problem.
> > > > As recognized by the reviewer, this goal is naturally captured by BCE loss whose negative value lower bounds the mutual information and is an intuitive proxy of the latter.
> > > > In experiments
> > > > **we evaluated how our attack can effectively achieve this goal empirically by reporting the decrease of BCE loss.**
> > > >
> > > > - We proposed using FLD score as a simple yet effective proxy for BCE loss minimization that helped make attack objective solvable.
> > > > In experiments **we validated how change of FLD score can approximate corresponding change of BCE loss** and this result is reported in Figure 4 of the appendix.
> > > >
> > > > - We provided a detailed analysis on how our new attack goal will affect popular fairness metric demographic parity (DP).
> > > > In experiments **we reported that our attack successfully increased of DP violation on all 16 experimental settings while baselines suffered from several severe failure cases**.
> > > > We agree that this result is of great importance and should be included in the main body. We will update the manuscript accordingly.
> > > >
> > > > - We derived the minimal number of poisoning samples needed to launch a successful attack. This analysis sheds light on defending against the attack (by maximizing our the lower bound) and in experiments **we verify that as suggested by the analysis, reducing batch size can indeed help reduce the attack effectiveness**.
> > > >
> > > > Besides these main contributions, other contributions of this work are also supported by experiments.
> > > >
> > > > - We used elastic-net penalty to regulate perturbation on poisoning samples to increase the stealthiness of our attack.
> > > > This is also verified by the reduction of perturbation norms in the experiments. Moreover, we showed that this penalty can conduct feature selection for the attack. The unselected features are considered as *robust features* and we believe that this provides a new view of understanding the vulnerability of existing FRL methods.

---

### Author Response · Authors · 2023-11-20
**Summary of revision on the paper, and we thank all reviewers for providing valuable feedbacks.**

Dear reviewers and Area Chairs:

We highly appreciate your time and effort have put into our paper. We have uploaded a revised draft with the following major updates. All revised texts have been colored in red.

1. To evaluate how different attacks affect the utility of victim models, we reported the accuracy of predicting label $y$ from learned representation $z$ under different portion of poisoning samples in the Appendix D.4. We did not find significant drop of prediction accuracy as shown in Figure 6 and Figure 17.

2. To evaluate the effectiveness of our attack, we further evaluate all attackers on COMPAS and Drug Consumption datasets, results (change of BCE loss, DP violations, and accuracy) are presented in Appendix D.8. Again our attack outperformed existing baselines to a large margin.

3. Some rewording and rephrasing of some less accurate statements.

Thanks in advance for your time, and we are looking forward to hearing from you if you have any additional questions or concerns.

Best regards,
Authors

---

### Meta-Review · Area_Chair_jg3Z · 2023-12-06

**Metareview:**

This paper focuses on data poisoning attacks against fair machine learning models, an important and interesting problem. The paper introduces an innovative information-theoretic approach to execute poisoning attacks in fair machine learning.

This novel framework is particularly noteworthy for its use of mutual information as a key metric, contrasting with conventional methods like demographic parity. The authors propose using Fisher's Linear Discriminant Analysis (FLD) as an efficient proxy for mutual information, also discussing the potential of other analytic proxies and comparing them with FLD.

The weakness includes the strong assumption in threat model (the attack assumes a white-box threat model) and the lack of explanation against fair representation learning (FRL) and why the attackers are interesting in making the model unfair.

**Justification For Why Not Higher Score:**

The weakness includes the strong assumption in threat model (the attack assumes a white-box threat model) and the lack of explanation against fair representation learning (FRL) and why the attackers are interesting in making the model unfair.

**Justification For Why Not Lower Score:**

This novel framework is particularly noteworthy for its use of mutual information as a key metric, contrasting with conventional methods like demographic parity. The authors propose using Fisher's Linear Discriminant Analysis (FLD) as an efficient proxy for mutual information, also discussing the potential of other analytic proxies and comparing them with FLD.

---

### Decision · Program_Chairs · 2024-01-16

Accept (poster)